



# Impact of wave-water level non-linear interactions for the projections of mean and extreme wave conditions along the coasts of western Europe

5  Alisée A. Chaigneau[1,2], Stéphane Law-Chune[2], Angélique Melet[2], Aurore Voldoire[1], Guillaume Reffray[2], Lotfi Aouf[3]

[1]CNRM UMR 3589, Météo-France/CNRS, Toulouse, France
[2]Mercator Ocean International, Toulouse, France
10  [3]Météo-France, Toulouse, France

*Correspondence to*: Alisée A. Chaigneau (achaigneau@mercator-ocean.fr)

**Abstract.** Wind-waves are a main driver of coastal environment changes. Wave setup and runup contribute to coastal hazards such as coastal flooding during extreme water level (EWL) events. Wave characteristics used to estimate wave setup are sensitive to changes in water depth in shallow waters. However, wind-waves models used for historical simulations and 15  projections typically do not account for water level changes whether from tides, storm surges, or long-term sea level rise. In this study, the sensitivity of projected changes in wind-wave characteristics to the non-linear interactions between wind-waves and water level changes is investigated along the Atlantic European coastline. For this purpose, a global wave model is dynamically downscaled over the northeastern Atlantic for the 1950-2100 period and for two climate change scenarios (SSP1-2.6 and SSP5-8.5). Twin experiments are performed by accounting (or not) for hourly variations of water level from 20  regional ocean simulations in the regional wave model. The largest impacts of wave-water level interactions are found in the Bay of Mont-Saint-Michel in France, due to a large tidal range of 10 m. At this location and during an historical extreme event, significant wave height was found to be up to 1 m higher (or +25 %) when considering water level variations, leading to an increase in wave setup by between +8.4 cm and +14.7 cm, depending on the value of the beach slope used. At the end of the 21[st] century under SSP5-8.5 scenario, the wave simulation including water level variations exhibits an increase in 25  extreme significant wave heights and wave setup values by up to +20 % and +10 % respectively. These results are found for many coastal points of the large continental shelf where shallow-water dynamics prevail, and especially so in macro-tidal areas.

## 1 Introduction

Coastal zones are among the most densely populated and urbanized areas in the world (McMichael et al., 2020; Neumann et 30  al., 2015; Wolff et al., 2020). Wind-waves are a major driver of coastal environment changes (Ranasinghe, 2016) and can drive coastal marine hazards such as coastal flooding (Melet et al., 2020b). Coastal marine flooding is most severe during




extreme water level (EWL) events. EWLs actually cause most of the sea level-related damages and are on the rise due to mean sea level rise (e.g. Fox-Kemper et al. 2021, Le Cozannet et al. 2022).

Wind-waves contribute to EWL events at the coast via wave setup and runup, combined with astronomical tides, storm
surges (due to low atmospheric surface pressure and wind setup) and mean sea level changes. Wave setup corresponds to the time-mean (over several wave groups) elevation of the water level in the shallow surf zone due to breaking waves. Depth-induced wave breaking releases the momentum transported by wind-waves, resulting in a shoreward decrease of the wave-momentum flux. The latter acts as a horizontal pressure force that tilts the sea surface, resulting in a time-mean elevation of the water level increasing from the breaking point to the shore (Dodet et al., 2019; Longuet-Higgins and Stewart, 1964).
Wave setup scales with wave characteristics such as the offshore wave height and wavelength (Holman, 1986; Stockdon et al., 2006; Dodet et al., 2019). As a rule of thumb, wave setup reaches 10 % to 20 % of the significant wave height at the breaking (Holman, 1986; Guza and Thornton, 1981). Wave setup can thus substantially contribute to EWLs at the coast during energetic wave conditions, with amplitude reaching several decimeters being reported (Pedreros et al., 2018; Guérin et al., 2018). Lavaud et al., 2020 also revealed that wave setup contributes significantly to storm surge peaks with values
reaching 40 % and 23 % at two different locations. Wave runup corresponds to the highest waterline elevation reached by individual waves and is of prime importance for overtopping of coastal defenses (Almar et al., 2021). In this study, wave contributions will be limited to wave setup.

To build knowledge on future changes in wind-wave climate, a growing number of global and regional wind-wave projections have been developed and intercompared (Hemer et al., 2013; Hemer and Wand 2017; Morim et al., 2018, 2021;
Lobeto et al., 2021; Meucci et al., 2020). Projections are commonly based on dynamical wave models forced by projected surface winds from general circulation models, notably from climate models contributing to the Coupled Model Intercomparison Project (CMIP), with potential downscaling of atmospheric forcing. Regional dynamic downscaling can be used to provide wind-wave projections at higher resolution. A multi-model analysis is required to assess uncertainties and robustness of projected wind-wave climate changes. Morim et al., 2018, 2019 provided a review of wind-wave projections.
Over the northeastern Atlantic and Mediterranean Sea bordering the coasts of western Europe, models project a robust decrease in annual and seasonal mean significant wave height, together with a decrease in the mean wave period. Regarding mean wave direction, a robust clockwise change is projected for the Atlantic Iberian coast. Extreme significant wave heights are also consistently projected to decrease over the northeastern Atlantic and Mediterranean Sea (Morim et al., 2018, 2021; Aarnes et al., 2017).

Wave characteristics used to estimate wave setup are sensitive to water level changes in shallow waters, where waves interact with the ocean bottom. However, wind-waves models used for historical simulations and projections typically do not account for water level changes, whether from tides, storm surges, or long-term sea level rise. Yet, wave statistics have been shown to be sensitive to sea level rise (Chini et al., 2010; Wandres et al., 2017; Arns et al., 2017) and to tide-surges during



extreme events (Alari, 2013; Viitak et al., 2016; Yu et al. 2017; Fortunato et al., 2017; Lewis et al., 2019; Staneva et al.,
2021). A review of tide and sea level rise effects on waves and wave setup is provided in Idier et al., 2019. Tides-wave setup
interactions can induce changes in total water level at the coast of a few centimeters to tens of centimeters in some cases
(Idier et al., 2019). Projections of wave setup during extreme events could thus differ substantially when interactions
between water level and waves are accounted for.

The present study aims at investigating the sensitivity of wind-wave mean and extreme characteristics to non-linear
interactions between wind-waves and water level changes. In particular, wave setup is presented as an indicator for coastal
hazard related to coastal flooding. To that aim, regional hindcasts and projections of wind-waves are produced over the
1950-2100 period considering two climate change scenarios corresponding to low-emissions, high-mitigation (SSP1-2.6) and
high-emissions, low-mitigation (SSP5-8.5) pathways (O'Neill et al., 2016). The simulations are produced over the
northeastern Atlantic, called the IBI domain (Iberian-Biscay-Ireland). To assess the sensitivity of wave projections to the
wave-water level non-linear interactions, the regional wave model is adapted to consider hourly variations of water level
from a regional ocean model described in Chaigneau et al., 2022 for the same IBI domain.

The paper is organized as follows. Global and regional wind-wave model configurations and simulations are presented in
Sect. 2. Regional simulations for the northeastern Atlantic domain are compared to observations over the historical period
and to previously published 21$^{st}$ century projections in Sect. 3, in terms of mean and extreme conditions. Sect. 4 provides an
assessment of the impact of including hourly water level changes on wind-wave characteristics and wave setup along the
European Atlantic coastlines. Finally, results are discussed in Sect. 5 and conclusions are drawn in Sect. 6.

## 2 Methods: models and simulations

### 2.1 Wind-wave model: MFWAM

The global and regional wave simulations are produced with the MFWAM wave model, a modified version of IFS
ECWAM-CY41R2 cycle (ECMWF, 2014) developed at Météo-France for their operational applications (Aouf and Lefèvre,
2015). Additions from Météo-France concern (i) the use and adjustment of the (Ardhuin et al., 2010) source term for wave
breaking and swell damping dissipation (ST4 physics, Ardhuin et al., 2010), (ii) the use of a Phillips spectrum tail to better
constrain the surface roughness for high-frequency waves.

The wind input source term is based on Bidlot et al., 2007 and has been improved by considering wave damping by surface
friction and sheltering effects for short waves. The non-linear source term uses the discrete interaction approximation (DIA)
approach developed by Hasselmann et al., 1985. MFWAM primarily aims at describing the open ocean sea states. As such,
coastal (depth-induced) breaking is not included in MFWAM, but a better wave propagation around islands, mostly in the





Pacific Ocean, is obtained thanks to an island obstructions scheme. Therefore, the wave setup contribution is not obtained

directly and needs to be calculated based on the model outputs.

Supported by the assimilation of satellite observations, MFWAM is successfully operated within the Copernicus Marine Service (https://marine.copernicus.eu/) to provide near-real time (analyses/forecasts) and multi-year (reanalysis/hindcasts) wave products over both the global ocean and the northeastern Atlantic corresponding to the region of interest in this study (called IBI for Iberia-Biscay-Ireland regional seas).

## 2.2 Wave climate projections and associated forcing fields


The aim of the present study is to investigate the sensitivity of historical and projected sea states for the IBI region coastlines to the non-linear interactions between wind-waves and water level changes, notably during extreme events. To this end, a regional dynamical downscaling of a global wave model (global MFWAM configuration referred to as CNRM-HR-WAV, Sect. 2.2.2) forced by a global climate model (CNRM-CM6-1-HR, Sect. 2.2.1) was implemented over the northeast Atlantic

region (Sect. 2.2.3). Regional wave simulations (regional MFWAM configuration referred to as IBI-CCS-WAV, Sect. 2.2.3) are forced by 1/12 ° hourly surface currents from a regionally downscaled ocean model (IBI-CCS for Iberian–Biscay–Ireland Climate Change Scenarios, Sect. 2.2.1), using the same CNRM-CM6-1-HR climate parent model. In addition, a twin regional wave configuration was set up to investigate wave-water level interactions by considering hourly water level outputs from IBI-CCS in the wave model (IBI-CCS-WAV_ssh, Sect. 2.2.4). Figure 1 describes the downscaling strategy and

links between the different models used in this study. All wave simulations described in the following sections (CNRM-HR-WAV Sect. 2.2.2, IBI-CCS-WAV Sect. 2.2.3, IBI-CCS-WAV_ssh Sect. 2.2.4) were performed over the historical period (1950-2014) and the 21st century (2015-2100) under the SSP1-2.6 and SSP5-8.5 climate change scenarios.

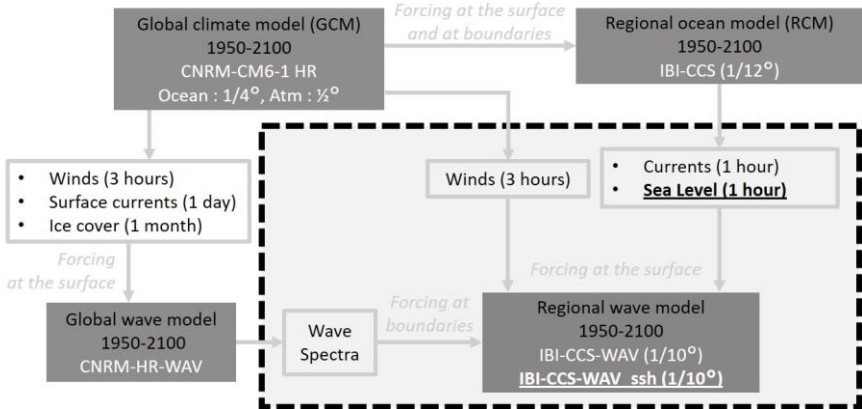

**Figure 1: Sketch of the downscaling strategy explaining the links between the different models used in this study.**



### 2.2.1 External forcing models: CNRM-CM6-1-HR global climate model and IBI-CCS regional ocean model

Our wave climate projections are driven by the CMIP6 CNRM-CM6-1-HR global climate model (GCM) simulations (Voldoire et al., 2019; Saint-Martin et al., 2021). The atmosphere and ocean components of CNRM-CM6-1-HR respectively have a 1/2 ° and 1/4 ° of horizontal resolution. The historical simulation of CNRM-CM6-1-HR is used over the 1950-2014 period. Over 2015-2100 period, simulations using the SSP1-2.6 and SSP5-8.5 climate change scenarios are used (O'Neill et al., 2016).

Regional wave projections are also forced by IBI-CCS, a regional ocean model at a 1/12 ° horizontal resolution. IBI-CCS was implemented in Chaigneau et al., 2022 to refine sea level projections of CNRM-CM6-1-HR over the northeastern Atlantic region using dynamical downscaling. For a more complete representation of processes driving coastal water level changes, tides and atmospheric surface pressure forcing are explicitly resolved in IBI-CCS in addition to the ocean general circulation (dynamic sea level).

### 2.2.2 Global wave simulations: CNRM-HR-WAV

Global wave simulations (CNRM-HR-WAV) are produced using MFWAM (Sect. 2.1) at a 1 ° resolution. CNRM-HR-WAV is forced by the three-hourly surface winds, monthly sea-ice cover and daily ocean surface currents taken from CNRM-CM6-1-HR corresponding simulations (Sect. 2.2.1). These global simulations provide the spectral wave information needed at the boundaries of the regional wave model described in Sect. 2.2.3 (Fig. 1).

CNRM-HR-WAV uses 2-min gridded global topography data from ETOPO2/NOAA (National Geophysical Data Center 2006). The model grid has a constant spacing in longitude but is compressed in latitude to maintain a constant resolution (Bidlot, 2012). A wind-wave growth calibration was performed to adjust the mean significant wave height of CNRM-HR-WAV to the Copernicus Marine Service WAVERYS wave reanalysis (Law-Chune et al., 2021) over the IBI domain. In our simulations, the wave spectrum is discretized in 24 directions and 30 frequencies starting from 0.035 up to 0.58 Hz. The time step is fixed at 720 s. Classical integrated wave parameters such as significant wave height (Hs) or average wave period (Tm) are generated three-hourly for CNRM-HR-WAV.

The aim of the study is not to characterize the uncertainties of wave projected changes over the IBI domain. However, we verified that CNRM-CM6-1-HR was consistent with other CMIP6 GCMs in particular in terms of extreme winds and extreme wind projections before using them to force the global and regional wave models. A comparison of extreme winds (99th percentile) between CNRM-CM6-1-HR, CMIP6 GCMs, the atmospheric reanalysis ERA5 (Hersbach et al., 2020) and wind observations from wave buoys (Wehde et al., 2021) is performed at different locations in the IBI region (Fig 2a). The three different locations considered (shown in Fig. 3) are chosen along storm trajectories in the northeastern Atlantic and North Sea (Lozano et al., 2004). Figure 2a shows that CNRM-CM6-1-HR is representative of an ensemble of 21 CMIP6



models over the historical period. In general, CNRM-CM6-1-HR is also in good agreement with ERA5 which is the

reference here. However, wave buoys observations seem to be significantly different from both GCMs and ERA5, except in

the North Sea. Figure 2b shows the projected changes for the extreme wind speed at the three locations. Projected changes in

extreme wind speed are quite small in all models and rather uncertain (large interquartile range). Projected changes are of the

same sign for 7, 9 and 10 models out of twelve for the three boxes respectively. In the English Channel and North Sea,

CNRM-CM6-1-HR shows an increase in extreme wind speed which is representative of the other CMIP6 models. In the

North Atlantic, CNRM-CM6-1-HR exhibits a large decrease in extreme wind speed which is in the range of the other GCMs

and of the same sign as most models.

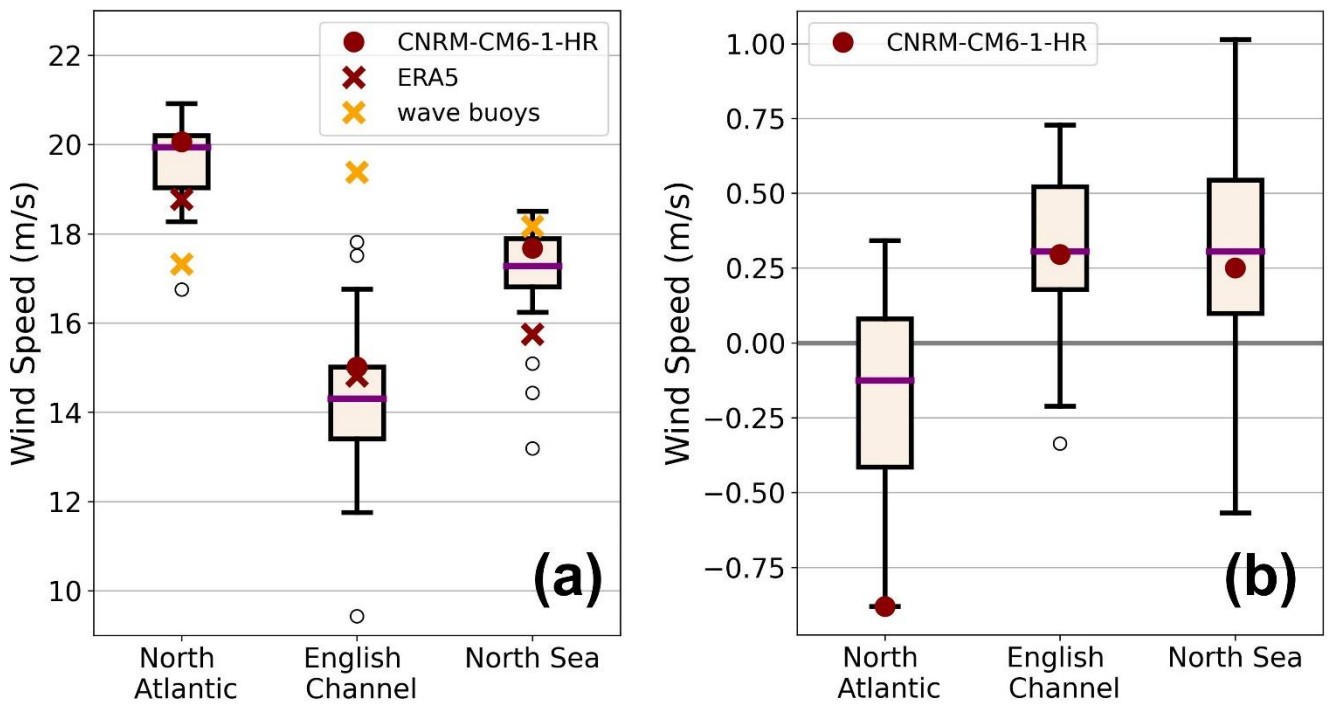

**Figure 2: (a) Extreme winds (99th percentile) for CNRM-CM6-1-HR (dark red dot), 21 different CMIP6 GCMs (black box), the**
**atmospheric reanalysis ERA5 (dark red cross) and wind observations from wave buoys (yellow cross) at the three locations in the**
**IBI region marked on Fig. 3a for the 1993-2014 period. The 2011-2022 period was chosen for the wave buoy in the North Sea as it**
**was the only period available. (b) Projected changes in extreme wind speed for CNRM-CM6-1-HR and 12 different CMIP6 GCMs**
**at the three locations marked on Fig. 3a under the SSP5-8.5 scenario (2081-2100 vs 1986-2005). The selected CMIP6 GCMs are**
**those with three-hourly atmospheric outputs. In (a) and (b), the purple line represents the median and the black box represents the**
**interquartile range. The black circles represent the outlier models i.e. models outside 1.5 times the interquartile range above the**
**and below the black box. Units are in m s$^{-1}$.**

**2.2.3 Regional wave model: IBI-CCS-WAV**

Regional wave simulations IBI-CCS-WAV (Iberian-Biscay-Ireland Climate Change Scenarios Wave) are produced using

MFWAM (Sect. 2.1) at a 1/10 ° resolution. The configuration was designed over the IBI domain based on a Copernicus



Marine Service (CMEMS) configuration (https://doi.org/10.48670/moi-00030). The regional domain covered by IBI-CCS-WAV extends from 27 to 61 ° N and 17 ° W to 8 ° E (Fig. 3), leading to a horizontal resolution ranging from 5.5 to 10 km. The IBI zone is interesting for wave modeling as it contains a variety of physical processes. First, the domain contains strong variations of bathymetry, with a wide continental shelf in the northern part of the domain (North Sea, English Channel) and a tight continental shelf in the southern part (Spain, Portugal, Morocco, Mediterranean Sea) (Fig. 3). There are also contrasting wave regimes: the Atlantic coasts are subject to very energetic events in terms of significant wave heights, wave periods and energy flows whereas the Mediterranean Sea and North Sea are more sheltered areas. In addition, the zone also contains very different tidal regimes with both macro and micro tidal regimes respectively in the English Channel/Celtic Sea and in the Mediterranean Sea.

**Figure 3: (a) Bathymetry (m) of the IBI domain in the regional wave model. The shelf break (defined by the 200 m isobath) is indicated by the solid yellow line. The yellow dotted lines indicate the areas where the waves start to interact with the bathymetry in IBI-CCS-WAV, identified when h<L/2 with h the bathymetry and L the mean wind-wave wavelength over the 1993-2014 period. The red dots represent the locations of the wave buoys from the Copernicus Marine Service (Wehde et al., 2021) used for the validation in Sect. 3.1. The three red boxes are used in Sect. 4 to validate extreme winds in CNRM-CM6-1-HR. Orange diamonds indicate the wave buoys used for the wave roses calculation of Sect. 3.1.2 and 3.2.2 (North Atlantic (NA) buoy 6200093 and Belle-Ile (BI) buoy 6200074; used for the extreme wind validation of Sect. 4 (North Atlantic (NA) buoy 6200093, English Channel (EC) buoy 6200103 and North Sea (NS) buoy 6200145). (b) Bathymetric adjustment (Sect. 2.2.4) corresponding here to the M2 tidal range from IBI-CCS (1993-2014). The lines indicate the areas where the waves start to interact with the bathymetry**



**at low tide (dashed white lines) and at high tide (solid yellow lines). The yellow diamonds indicate the zones where the impact of including hourly water level outputs in the wave model is assessed in Sect. 4.**

IBI-CCS-WAV is used to dynamically downscale the CNRM-HR-WAV global simulations described in Sect. 2.2.2. The dynamical downscaling method allows the resolution of regional processes at a finer scale. The method consists in forcing the regional model (IBI-CCS-WAV, 1/10 °) at its lateral boundaries (3-hourly wave spectra) by the larger scale wave model (CNRM-HR-WAV, Sect 2.2.2) and at the ocean surface (3-hourly surface winds, hourly currents) by the CNRM-CM6-1-HR climate model and the regional ocean model IBI-CCS (Sect. 2.2.1). In the regional wave model, the bathymetry differs from

the global model with the use of a smoothed ETOPO1 ocean bathymetry (https://sos.noaa.gov/datasets/etopo1-topography-and-bathymetry/). The spectral discretization is the same as for the global model CNRM-HR-WAV. The time step of IBI-CCS-WAV is set to 240 s. Classical integrated wave parameters are generated hourly for IBI-CCS-WAV.

### 2.2.4 Inclusion of water level variations in the regional wave model: IBI-CCS-WAV_ssh

The water levels over which the waves propagate control the shallow water wave dynamics and the refraction by bathymetric

gradients, as well as bottom friction induced wave breaking (which is not relevant here given the resolution of the wave models).

To measure the impact of wave-water level interactions in the IBI region, a twin configuration to IBI-CCS-WAV (Sect. 2.2.3) was set up to consider water level variations as an additional forcing: IBI-CCS-WAV_ssh. For this purpose, MFWAM has been modified to include an hourly water level forcing (coming from IBI-CCS simulations, Chaigneau et al., 2022, Fig.

1) which adjusts bathymetric depth (Fig. 3b) to hydrodynamic effects such as tides, storm surges, dynamic sea level associated with ocean circulations but also to the long-term mean sea level rise over the next century. In practice, water elevation is read at the same time as ocean currents (Fig. 1) and added to the topographic depth at each forcing time step. Wave propagation parameters in shallow water such as group velocities and wave numbers are tabulated at the beginning of the simulation according to a fixed list of depths and frequencies in the form of look up tables. The depth discretization for

the propagative aspects was adapted with a first level at 3 m and a vertical resolution of the order of 15 cm near the surface. A minimum time-mean water depth of 6 m was chosen to be consistent with that applied in the ocean simulation from IBI-CCS (Chaigneau et al., 2022). This value avoids the occurrence of uncovered banks in macro-tidal areas, especially around Mont Saint Michel in France. These settings are applied for all regional wave simulations, whether they are forced by time-varying water levels (IBI-CCS-WAV_ssh) or not (IBI-CCS-WAV).

### 2.3 Wave setup calculation

The present study aims at investigating the sensitivity of projected changes in mean and extreme wave conditions to waves-sea level non-linear interactions. In particular, the focus is given on the wave setup which drive coastal sea level hazards such as coastal flooding. Wave setup and runup can be computed with wave-resolving coastal models such as Xbeach





(Roelvink et al., 2009), SWASH (Zijlema et al., 2011) or BOSZ (Pinault et al., 2020). Such models require a high resolution

of several meters and nearshore profiles as inputs. They cannot yet simulate nearshore morphological changes over long time periods and at large spatial scales due to their limitations to represent cross-shore sediment exchanges (Elsayed and Oumeraci, 2017).

Therefore, at first order, wave setup and runup can be predicted via empirical formulations that relate them to a set of simple environmental parameters (Dodet et al., 2019). Limitations related to the use of parameterizations have been extensively

discussed in Melet et al., 2020 with a sensitivity analysis of projected wave setup and runup changes to the choice of empirical formulations. In our study, the wave contribution to sea level is limited to wave setup. As the aim is to provide a first order estimate of wave setup projected changes, wave setup estimates are based on an empirical formulation (Stockdon et al., 2006).

Wave setup has been shown to strongly depend on the foreshore beach slope $\beta$. For sandy beaches, a commonly used

empirical formulation to compute the wave setup ($\eta$) accounting for $\beta$ is:

$$\eta = 0.35\beta\sqrt{H_s L_p} \quad (1) \text{ (Stockdon et al., 2006)}$$

where $H_s$ is the deep-water significant wave height, $L_p$ is the deep-water peak wave wavelength related to the deep-water peak wave period $T_p$ through the deep-water linear dispersion relationship: $L_p = \frac{g}{2\pi}T_p{}^2$ ; $g$ is the acceleration of gravity. As explained in Melet et al., 2020, the foreshore beach slope evolves over different time scales (extreme events, seasonal,

interannual, and in response to sea level rise) and spatial scales (from alongshore at a given local beach to regional scales) and generally ranges between 0.01 and 0.20 (Komar, 1998). At the moment, however, no observations of the foreshore beach slope applicable in empirical formulations is available worldwide or in Europe. Therefore, a time and space constant beach slope value is commonly used for global and regional studies (Serafin et al., 2017; Melet et al., 2018, 2020a). In Melet al., 2020, a beach slope of 0.04 is used, corresponding to the median value of local values at 308 sites along the global ocean

coastlines. In Serafin et al., 2017, a constant beach slope of 0.05 is used corresponding to the regional mean based on observations. In Vos et al., 2020, regional beach slopes are provided for the coasts of southeastern Australia and USA California based on satellite data. The spatial mean of these regional estimates is 0.06-0.07.

Given the spatial inhomogeneity of $\beta$ and its small scale nature, simpler formulations of wave setup that do not depend on $\beta$ but rather on the significant wave height have been proposed for use in regional scale studies, for example : $\eta = 0.2H_s$ (2)

(Holman, 1986; Vousdoukas et al., 2017, 2018) or $\eta = 0.16H_s$ (Atkinson et al. 2017). The formulation (2) however can lead to overestimations of the wave setup contribution as the applicability range is for beach slopes between 0.07 and 0.2 (Holman et al., 1986). Therefore, we chose not to use this formulation in this study.



In this study, for the wave setup changes of Sect. 3.2 and 4, we provided a wave setup scaling $\Delta\sqrt{H_s L_p}$ rather than using the formulation of equation (1) to allow our results to be scaled with different beach slopes or empirical formulae (Dodet et al., 2019). To provide a range of wave setup changes estimates, formulation (1) was used with beach slopes of 0.04 and 0.07, which correspond to the low and high spatial-mean values found in previous broad-scale studies (Melet et al., 2020a, Vos et al., 2020).

## 3 Validation and projections of IBI-CCS-WAV, without waves-sea level interactions

### 3.1 Validation of IBI-CCS-WAV over the 1993-2014 period

IBI-CCS-WAV is validated over the 1993-2014 period against Copernicus Marine Service products: a regional wave reanalysis which will be referred to as IBI-WAV thereafter (García San Martín et al., 2021; Toledano et al., 2021) and observations from wave buoys (Wehde et al., 2021). The IBI-WAV reanalysis covers the whole 1993-2020 period and has a horizontal resolution of 5 km. IBI-WAV uses the currents from the IBIRYS regional ocean reanalysis (Levier et al., 2020). In the present study, we considered the IBI-WAV reanalysis as the reference for the domain as Toledano et al., 2021 have shown that its performance was good compared to satellite and buoy observations over the 1993-2019 period. The selected wave buoys have a temporal data coverage of at least 60% over the validation period. The 1993-2014 period was chosen for the validation period because it corresponds to the intersection between the period covered by the IBI-WAV regional reanalysis (starting in 1993) and the historical period of IBI-CCS-WAV (ending in 2014).

The ability of IBI-CCS-WAV and IBI-CCS-WAV_ssh to reproduce observed distributions is assessed for the mean state and the 99[th] percentile of the significant wave height and peak period since these variables are then used to compute the wave setup scaling (Sect. 3.2, 4).

### 3.1.1 Significant wave height and mean wave period

**Mean state validation**







**Figure 4: (a), (b), (c) and (d) show the mean significant wave height (Hs, in m) over the 1993-2014 period for: (a) IBI-CCS-WAV. (b) Differences between IBI-CCS-WAV and the reanalysis IBI-WAV. (c) Bias between IBI-CCS-WAV and CMEMS wave buoys. (d) Scatter plot at each wave buoy location of simulations IBI-CCS-WAV (yellow marks) and IBI-WAV reanalysis (blue marks) vs observations. (e), (f), (g) and (h) are the corresponding figures for peak period (Tp, in s). Note the different color bars in (a) and (b), (c), and in (e) and (f), (g). For (b) and (f) the domain is limited to the domain distributed by the Copernicus Marine Service, with a cut in the northern part. In (d) and (h), the thin dashed lines indicate the 20 % error margin. The RMSE is calculated as the root mean squared deviations from the line y=x (spatial RMSE).**

Figure 4 illustrates the mean state validation of the IBI-CCS-WAV simulation against the IBI-WAV reanalysis and observations from wave buoys. In general, the mean significant wave height and mean peak period of IBI-CCS-WAV seem to be in reasonable agreement with both the reanalysis IBI-WAV and the wave buoys over the 1993-2014 period. Around 40 °N in the deep ocean, IBI-CCS-WAV nevertheless exhibits a positive bias for the significant wave height inherited from the global wave model forcing through the boundaries (Fig. 4b). This feature is due to the wind forcing taken from the GCM. In CNRM-CM6-1-HR, the westerlies are slightly shifted southward. As a result, significant wave heights are slightly overestimated in the IBI-CCS-WAV southern domain and underestimated in the northern part of the domain, around Ireland, leading to an overall relative error of 10 %. In the deep ocean, the peak period of IBI-CCS-WAV seems to be in good agreement with the reanalysis IBI-WAV (Fig. 4f). However, the significant wave height and peak period differences




between IBI-CCS-WAV and IBI-WAV are often larger in coastal zones and can reach a relative error of 20 % in the Gulf of Cadiz (Fig. 3a,b,e and f). Differences in coastal zones mainly arise from the different surface currents forcings, coming from IBI-CCS for IBI-CCS-WAV and from IBIRYS (Levier et al., 2020) for IBI-WAV. The currents are particularly different around the Strait of Gibraltar. At this location, the incoming transport and the currents from IBIRYS are more intense than

285    those of IBI-CCS. Indeed, IBI-CCS has been corrected in Chaigneau et al. 2022 to obtain a more accurate transport through the Strait of Gibraltar.

Figure 4c displays the mean significant wave height biases between IBI-CCS-WAV and wave buoys over the 1993-2014 period. The spatial pattern of the biases is generally in agreement with Figure 4b showing a negative bias in the northern part of the domain and a positive bias around the Strait of Gibraltar (Fig. 4b and c). Around the Iberian Peninsula, the biases

290    found for the peak period between IBI-CCS-WAV and the reanalysis IBI-WAV seem to be in contradiction with those found between IBI-CCS-WAV and the wave buoys (Fig. 4f and g). Toledano et al., 2021 also reported large errors in northern Spain between IBI-WAV and the wave buoys for the mean wave period. The uncertainty seems to be large in this region and IBI-CCS-WAV is within the range of uncertainties. The scatter plots of Figure 4d and h show that the performances of both IBI-CCS-WAV and IBI-WAV are quite similar on average over the domain with a root mean square error (RMSE) of the

295    same order of magnitude: about 20 cm for the significant wave height and 1 s for the peak period. Figure 4h must be interpreted with caution as the observations of peak period are scarce and located close to the coast. Most likely coastal buoys are subject to very local effects that are poorly represented at the 1/10 ° model resolution (Fig 4g,h).

**Extreme validation: 99[th] percentile**





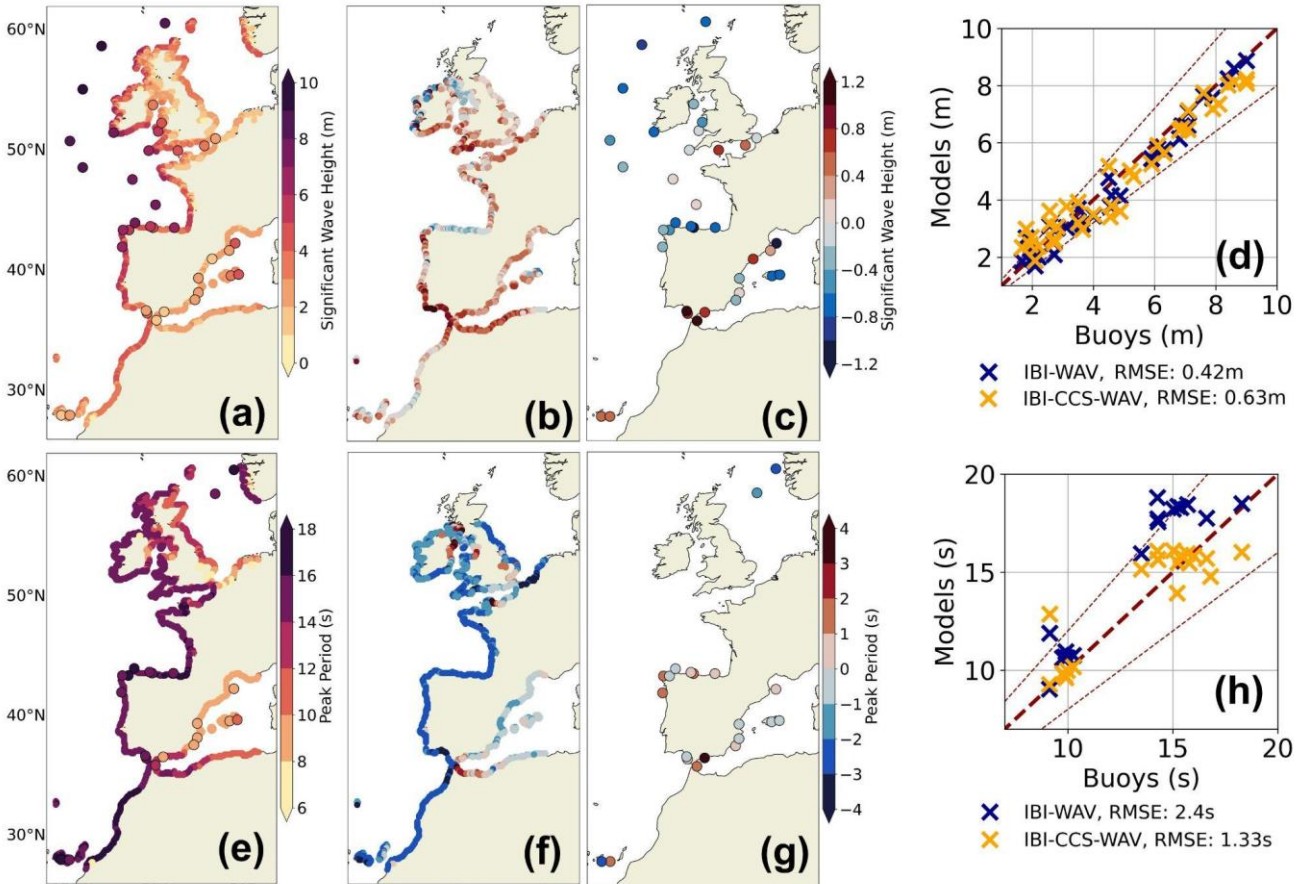

**Figure 5: (a), (b), (c) and (d) show the 99[th] percentile (based on hourly outputs) significant wave height (Hs, in m) over the 1993-2014 period for: (a) IBI-CCS-WAV. (b) Differences between IBI-CCS-WAV and the reanalysis IBI-WAV. (c) Bias between IBI-CCS-WAV and CMEMS wave buoys. (d) Scatter plot at each wave buoy location of simulations IBI-CCS-WAV (yellow marks) and IBI-WAV reanalysis (blue marks) vs observations. (e), (f), (g) and (h) are the corresponding figures for peak period (Tp, in s). Note the different color bars in (a) and (b), (c), and in (e) and (f), (g). For (b) and (f) the domain is limited to the domain distributed by the Copernicus Marine Service, with a cut in the northern part. In (d) and (h), the thin dashed lines indicate the 20% error margin. The RMSE is calculated as the root mean squared deviations from the line y=x (spatial RMSE). Note that the color scales for the biases are larger than for Figure 4.**

Figure 5 illustrates the validation of extremes (99th percentile) in the IBI-CCS-WAV simulation. The scatter plots of Figure 5d and 5h show that IBI-CCS-WAV satisfactorily reproduces extreme significant wave heights and peak periods, with an overall relative error of 14 % and 9 % for Hs and Tp, respectively. These values are comparable in value to the relative error of 13 %, 18 % and 20 % found in Lobeto et al., 2021 for the 5-, 20- and 50-year return periods of significant wave height.

The performances of both IBI-CCS-WAV and IBI-WAV are close, with a slight underestimation of the largest extreme significant wave heights in both models. In addition, IBI-CCS-WAV seems to overestimate the smallest 99th percentile of significant wave height, particularly in the Gulf of Cadiz and around the Strait of Gibraltar where the values are 1 m too large (Figure 5b and c). This feature is also mainly associated with the currents which are quite different around the Strait of





Gibraltar in IBI-CCS-WAV compared to those of IBI-WAV due to the complexity of the zone. Figure 5f shows the comparison of the peak period 99[th] percentile between IBI-WAV and IBI-CCS-WAV. Differences of 3 s (relative error of 20 %) are found between IBI-CCS-WAV and IBI-WAV along the Atlantic coasts (Fig. 5f). However, this feature seems to be related to an overestimation of the extreme peak periods in the reanalysis IBI-WAV as the differences do not appear when
IBI-CCS-WAV is directly compared to wave buoys (Fig. 5g). The scatter plot of Figure 5h also shows a smaller RMSE with IBI-CCS-WAV than with IBI-WAV when compared to wave buoys. This overestimation is also reported in Toledano et al., 2021 in which the reanalysis IBI-WAV is validated.

**3.1.2 Wave roses**

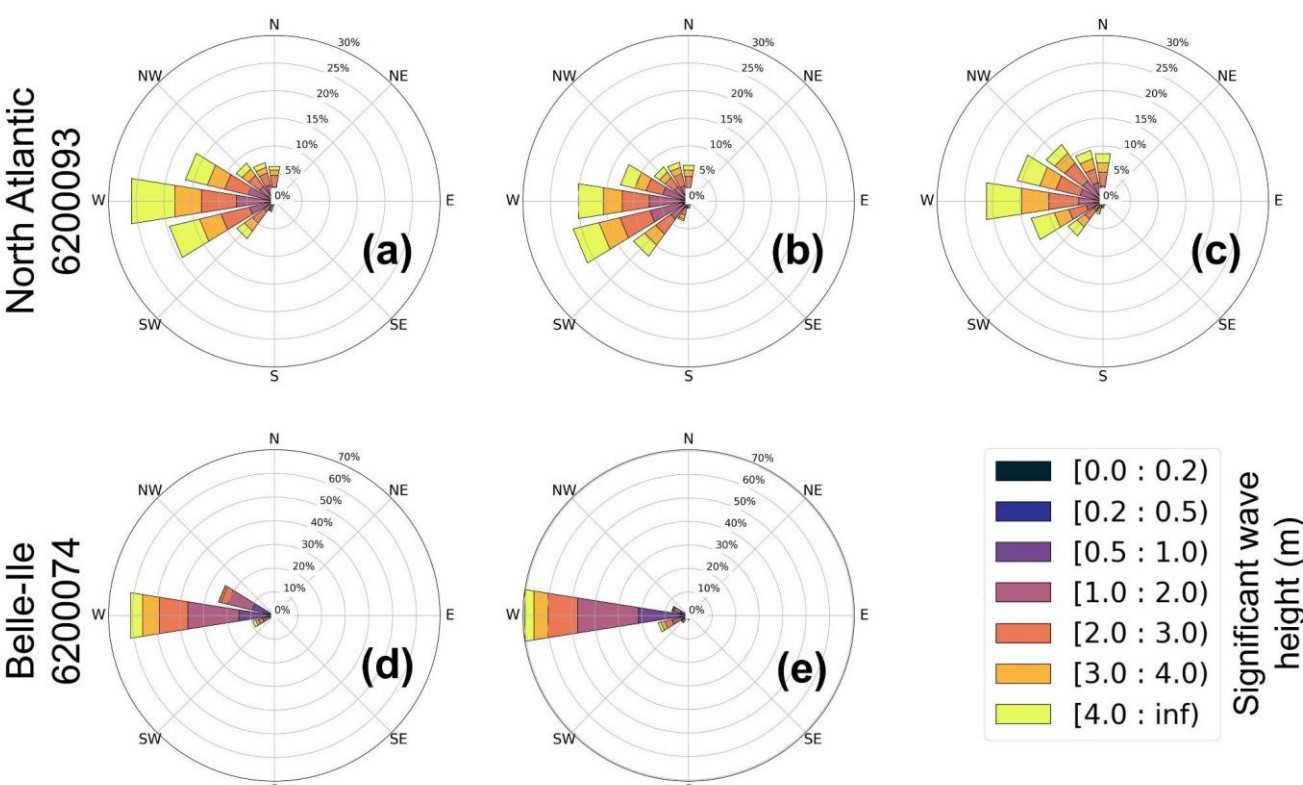

**Figure 6: Directional distributions of significant wave height at star locations of Fig. 3a: North Atlantic buoy 6200093 (first row), Belle-Ile buoy 6200074 (French Atlantic coast, second row). First column (a,d) are the wave roses based on wave buoy data over the 2003-2022 period for (a) and 2005-2022 for (d). Second column (b,e) are the roses for IBI-CCS-WAV over the 1993-2014 period and (c) for the IBI-WAV reanalysis over the 1993-2014 period. Different periods are chosen for the wave buoys because of the lack of data for the wave direction over the 1993-2014 period. Wave roses at North Atlantic buoy 6200093 location were**
**computed using mean wave direction. Wave roses at Belle-Ile buoy 6200074 location were computed using the wave direction at spectral peak as it was the data provided by the wave buoy. However, this variable was not an output of the IBI-WAV reanalysis. Colors indicate the wave height distribution in each direction bin.**

Directional distributions are shown on wave roses at two locations in the Atlantic Ocean (Fig. 3a). As both locations are found in the westerlies-exposed Atlantic Ocean, the wave roses indicate dominant waves in the west, west-northwest and



west-southwest directions. For North Atlantic buoy 6200093, IBI-WAV shows the same main west direction as the buoy data (Fig. 6a and c). However, the observed west distribution is slightly underestimated (by 5 %) by IBI-CCS-WAV and IBI-WAV, especially for the highest waves, superior to 4 m. In IBI-WAV, this could be associated with a larger directional spread of the biggest waves coming from the north (Fig. 6c). In IBI-CCS-WAV, waves tend to have a slight southward direction bias, as the dominant wave direction is west-southwest (Fig. 6b). For Belle-Ile buoy 6200074, the west direction

represents 70 % of occurrence in IBI-CCS-WAV against 60 % for the wave buoy (Fig. 6d and e). Half of the 70 % are related to waves with a significant wave height of less than 2 m whereas for the buoy it represents only 25 %. This difference of 25 % is found in the west-northwest direction bin for the wave buoy data.

Overall, the IBI-CCS-WAV regional wave model has shown good performances compared to the IBI-WAV reanalysis and wave buoys, although observations are scarce. For the future period, as in the present study we use a single GCM forcing, we

assess the regional projections compared to previous published studies.

## 3.2 Regional wave projections under two climate change scenarios: SSP5-8.5 and SSP1-2.6

Regional projections over 2015-2100 are now presented for the SSP5-8.5 and SSP1-2.6 scenarios for the significant wave height and peak period validated in Sect. 3.1 and for wave setup scaling $\Delta\sqrt{H_s L_p}$ (Sect. 2.3).

### 3.2.1 Projected changes in significant wave height, peak period and wave setup scaling

**Mean state projections**





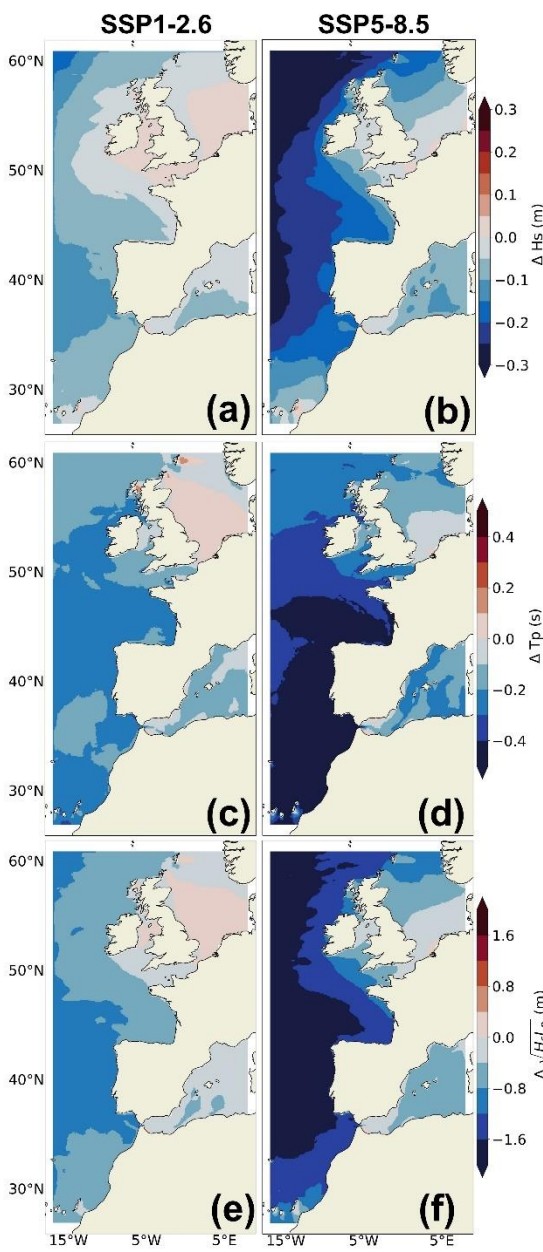

**Figure 7: Projected changes in incoming waves conditions from 1986-2005 to 2081-2100 under the SSP1-2.6 (first column) and SSP5-8.5 (second column) scenarios for (a,b) significant wave height (ΔHs, in m), (c,d) peak period (ΔTp, in s) and (e,f) scaling for wave setup ($\Delta\sqrt{H_s L_p}$, in m).**

Figure 7 illustrates projected changes in the significant wave height, peak period and wave setup scaling for the end of the century under two climate change scenarios. Projected changes under the SSP5-8.5 scenario are consistent with other studies





(Lobeto et al., 2021; Melet et al., 2020a; Morim et al., 2019; Aarnes et al., 2017; Casas-Prat et al., 2018) with a large decrease in the significant wave height, peak period and thus in the wave setup scaling in the Atlantic Ocean and Mediterranean Sea (Fig. 7b,d,e). For the SSP5-8.5 scenario, in the south of the domain, projected changes in the peak period

can reach -0.5 s which represents a decrease of 6 % in comparison to the historical period of Figure 3e. For the significant wave height, projected changes can reach -30 cm or -10 %. Changes in the wave height and peak period result from changes in the wave spectrum composed by different wave regimes (e.g. swells and wind-sea waves). The large decrease in the significant wave height under the SSP5-8.5 scenario (Fig. 7b) is due to a general decline in the wind speed forcing from CNRM-CM6-1-HR over the domain and in the North Atlantic Ocean, inducing changes in both wind-sea waves and swells

in the domain (not shown). The decrease in the wind speed under the SSP5-8.5 scenario is consistent with other CMIP6 GCM projections (Carvalho et al., 2021). In terms of wave setup scaling, projected changes can reach a decrease of 2 m or 6 % under SSP5-8.5 scenario along the north Iberian coasts (Fig. 7f). In terms of sea level equivalent, using parameterization (1) with a beach slope of 4 % and 7 % (Sect. 2.3), the decrease in the wave setup in the deep ocean can reach between -2.8 cm and -4.9 cm. These changes are thus rather small (but not negligible) compared to the projected sea level rise of about

+80 cm over the northeastern Atlantic domain (Chaigneau et al., 2022) but are of opposite sign. These changes in wave setup are mainly related to moderate changes in the peak period which is squared in formulation (1): 63 % of the wave setup changes are due to those in peak period and 37 % are due to changes in significant wave height. Except in the North Sea, the spatial patterns of the wave projected changes under the SSP1-2.6 scenario (Fig. 7a,c,e) are broadly the same as under the SSP5-8.5 scenario, but with an overall smaller magnitude.

**Extreme projections: 99[th] percentile**





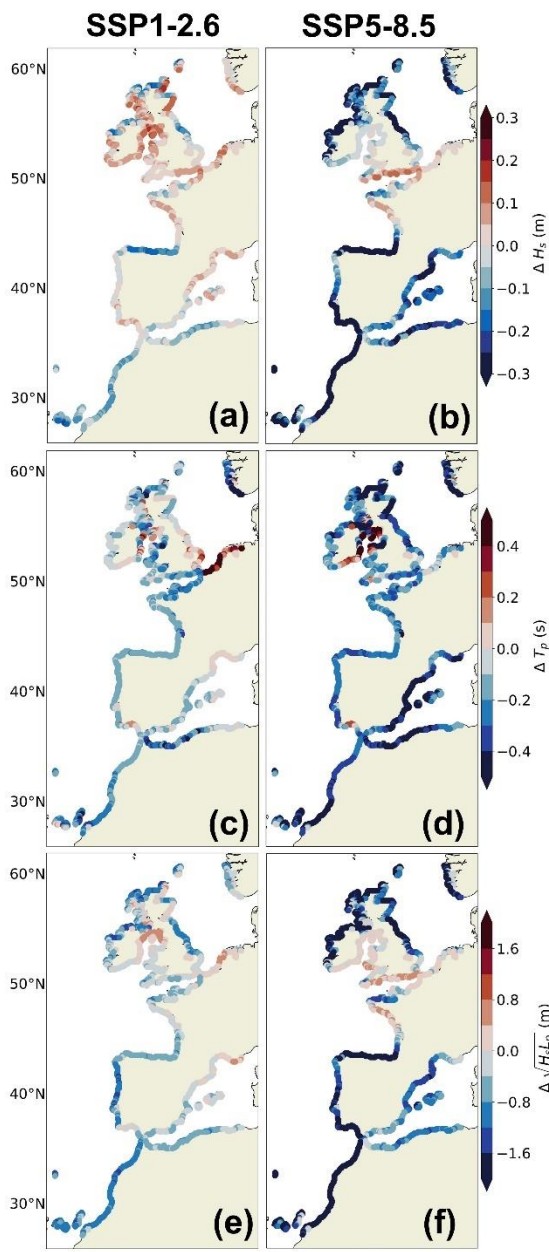

**Figure 8: Projected changes in incoming extreme (99ᵗʰ percentile) waves conditions from 1986-2005 to 2081-2100 under the SSP1-2.6 (first column) and SSP5-8.5 (second column) scenarios in IBI-CCS-WAV for (a,b) significant wave height (ΔHs, in m), (c,d) peak period (ΔTp, in s) and (e,f) scaling for wave setup (Δ$\sqrt{H_s L_p}$, in m).**

Figure 8 illustrates projected changes in the 99ᵗʰ percentile of significant wave height, peak period and wave setup scaling for

the end of the century under two climate change scenarios. Changes in the 99ᵗʰ percentile of peak period are moderate as it





generally represents a decrease of less than 2.5 % for the SSP5-8.5 scenario (Fig. 8c,d). For the significant wave height, projected changes in the 99<sup>th</sup> percentile under SSP5-8.5 scenario are large with a decrease of more than 30 cm or 12 % in the southern part of the domain. For both scenarios, projected changes in the 99<sup>th</sup> percentile of significant wave height are quite

different from those in the mean state, as reported in Morim et al., 2018. This is associated with different changes in the extreme wind speed forcing compared to those in the mean state (not shown). For example, for the SSP5-8.5 scenario, the large decrease in the extreme wind speed (Fig. 2b) and thus in the significant wave height is located in the North Atlantic south of 45 °N and north of 55 °N (Fig. 7b). This is consistent with other studies (Aarnes al., 2017; Meucci et al., 2021) in which the largest decrease in the 99<sup>th</sup> percentile of significant wave height is also found in the southern domain (Fig. 8b). In

the English Channel, Celtic Sea and French Atlantic coasts, the model even exhibits an increase in the extreme significant wave height that has not been reported in other studies for both scenarios (Fig. 8a,b). This increase is however consistent with projected changes in extreme wind speed shown in Fig. 2b for the English Channel. In the Mediterranean Sea, the SSP5-8.5 and SSP1-2.6 scenarios exhibit significant wave height projected changes of a different sign (Fig. 8a,b) associated with different projected changes in extreme winds (not shown). Over the whole domain, as projected changes in extreme

peak periods are small, projected changes in extreme wave setup scaling are mostly governed by those in extreme significant wave height (Fig. 8e,f) in contrast to the projected changes in mean state of Figure 7. For instance, along the north Iberian coasts, projected changes in wave setup scaling show a decrease of 2 m or 4 %, with 70 % of the changes due to those in significant wave height and 30 % due to changes in peak period. In terms of sea level equivalent, the decrease in extreme wave setup can reach between -2.8 cm and -4.9 cm using (1), as for the mean state of Figure 7.



### 3.2.2 Projected changes in wave roses

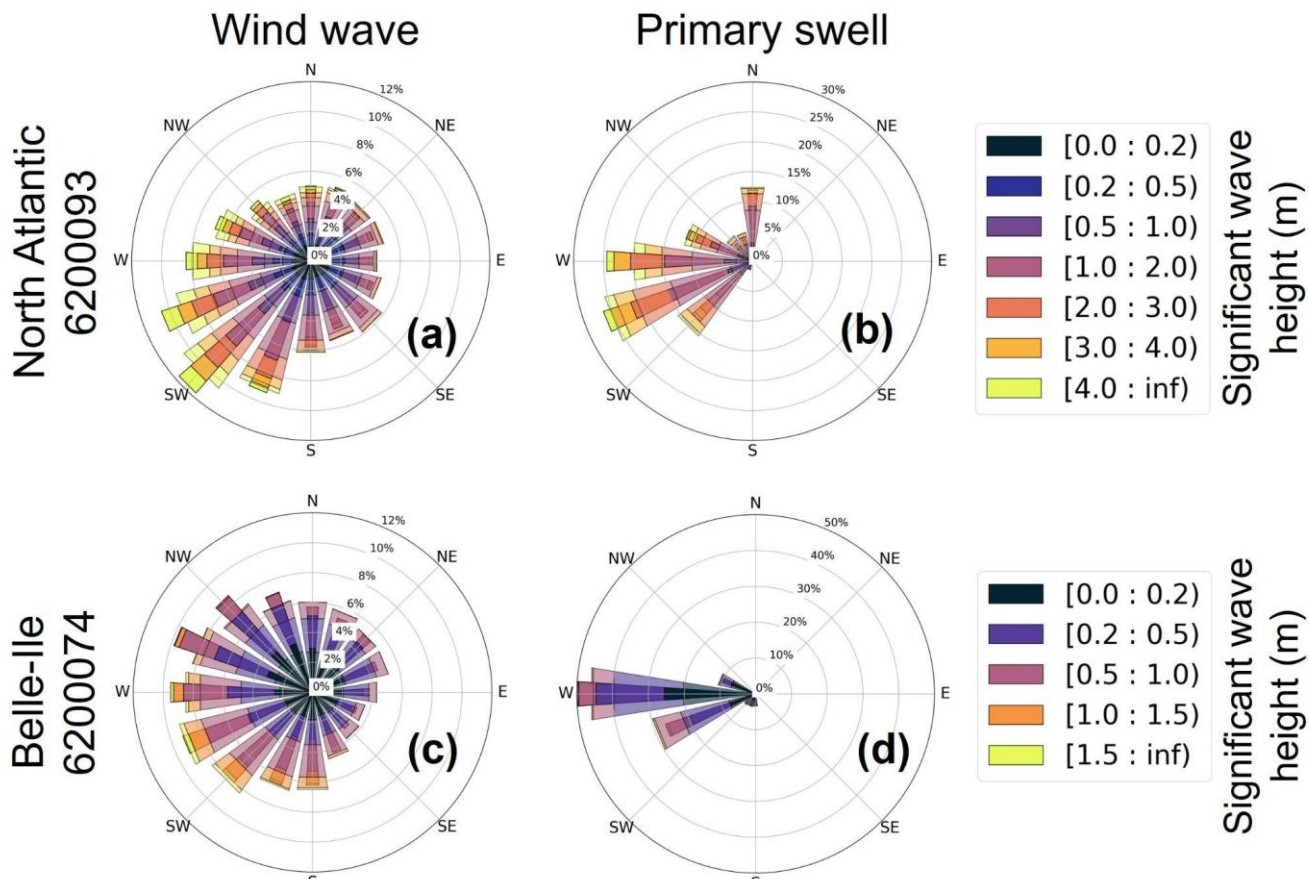

**Figure 9: Projected changes (SSP5-8.5 scenario) in directional distribution of significant wave height from 1986-2005 (wide angle bins, pale colors) to 2081-2100 (narrow angle bins, dark colors) in IBI-CCS-WAV at star locations of Fig. 3a: North Atlantic buoy 6200093 (first row), Belle-Ile buoy 6200074 (second row). The significant wave height and mean wave direction have been classified according to their origin: wind wave (a,c) and primary swell (b,d). Colors indicate wave height distribution in each direction bin.**

Projected changes in the directional distributions are shown on wave roses at the two locations validated in Figure 6. The wave roses have been decomposed into the wind wave and primary swell contributions. The significant wave height roses due to the primary swell contribution (Fig. 9b,d) are very close to the significant wave height roses of Fig. 6, showing that most waves are due to the primary swell at this location. At Belle-Ile buoy 6200074, the wave rose exhibits a change in the main direction of the wind-sea under the SSP5-8.5 scenario (Fig. 9c). Over the historical period, the main direction is west-southwest whereas it would be shifted clockwise to west-northwest direction (20 °) at the end of the century under the SSP5-8.5 scenario (Fig. 9c). In this zone, a clockwise shift in the wave direction has already been documented in Morim et al., 2019. This shift seems to come mainly from small waves with significant wave height less than 50 cm. For the primary swell at Belle-Ile, we observe a slight strengthening of the swell from the west direction, and thus a reduction of the wave




components coming from the southwest (Fig. 9d). The results are quite different for North Atlantic buoy 6200093. Projected changes exhibit a slight strengthening of the wind-sea wave heights in the southwest, west-southwest direction bins with an occurrence increased by a few percent (Fig. 9a). Also, for North Atlantic buoy 6200093, the projected changes in the primary swell are larger than at Belle-Ile buoy 6200074. The occurrence of the west direction has increased by 5 % (Fig. 9b).

Finally, a slight decrease in swells with significant wave heights of more than 4m is found under the SSP5-8.5 scenario for the west and west-northwest directions (Fig. 9b).

In Sect. 3.2, we verified that the projected changes of IBI-CCS-WAV were consistent with other studies. We notably observed a general decrease in mean and extreme significant wave height and peak period over the domain and a clockwise mean wave direction change along the French Atlantic coasts. IBI-CCS-WAV can therefore be used to assess questions

related to processes in a changing climate such as the influence of hourly water level variations on the wave conditions, particularly during extreme events.

## 4   Impact of non-linear wave-water level interactions accounted for in the regional wave model

### 4.1 Impact at two locations: Bay of Mont-Saint-Michel and French Atlantic coast

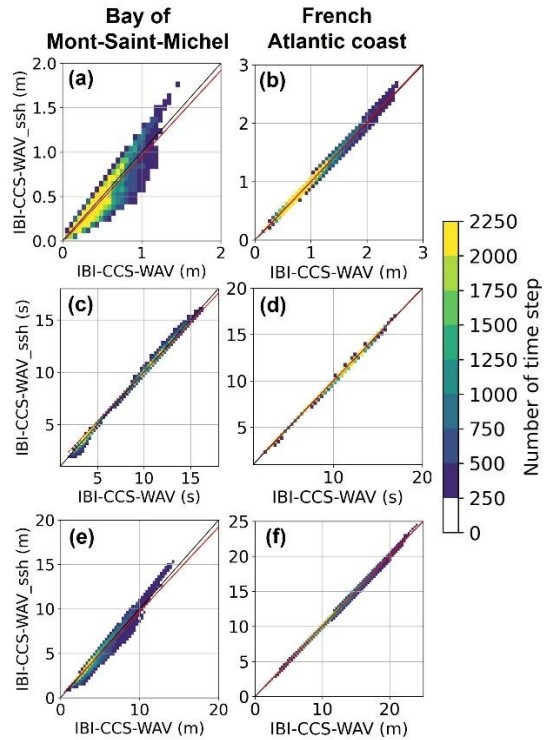

**Figure 10: Scatter plot of incoming wave conditions between IBI-CCS-WAV and IBI-CCS-WAV_ssh for the Bay of Mont-Saint-Michel (first column) and for the French Atlantic coast (second column) based on hourly outputs over the historical period 1993-**



**2014. The two locations are marked on Fig. 3b. The rows show 3 variables: (a,b) significant wave height (Hs, in m), (c,d) peak period (Tp, in s) and (e,f) scaling for wave setup ($\sqrt{H_s L_p}$, in m). The black line represents the y=x line and the red line is for the regression of the data.**

Figure 10 shows scatter plots of significant wave height, peak period and wave setup scaling between IBI-CCS-WAV and IBI-CCS-WAV_ssh at two macro-tidal locations over the historical period. In the Bay of Mont-Saint-Michel, strong hourly water level variations occur due to the large tidal range in the region (about 10 m, Fig. 3b). Therefore, in the Bay of Mont-Saint-Michel, the inclusion of hourly water level variations in the wave model, and thus of the tidal variations, has a substantial impact on the representation of significant wave heights (Fig. 10a). This is especially the case for the extreme
heights with an increase of about +20 % of the significant wave height (Fig. 10a). The inclusion of hourly water level variations in the wave model has globally not a large impact on the peak period. For the wave setup scaling, the higher impact of the hourly water level variations is found during extreme events as for the significant wave height (Fig. 10e). For the French Atlantic coast, the tidal range is large (4m, Fig. 3b) but smaller than in the Bay of Mont-Saint-Michel. The impact of including hourly water level variations on the significant wave height and thus on the wave setup scaling is less important,
but nevertheless follows the trends previously observed (Fig. 10b,d,f).

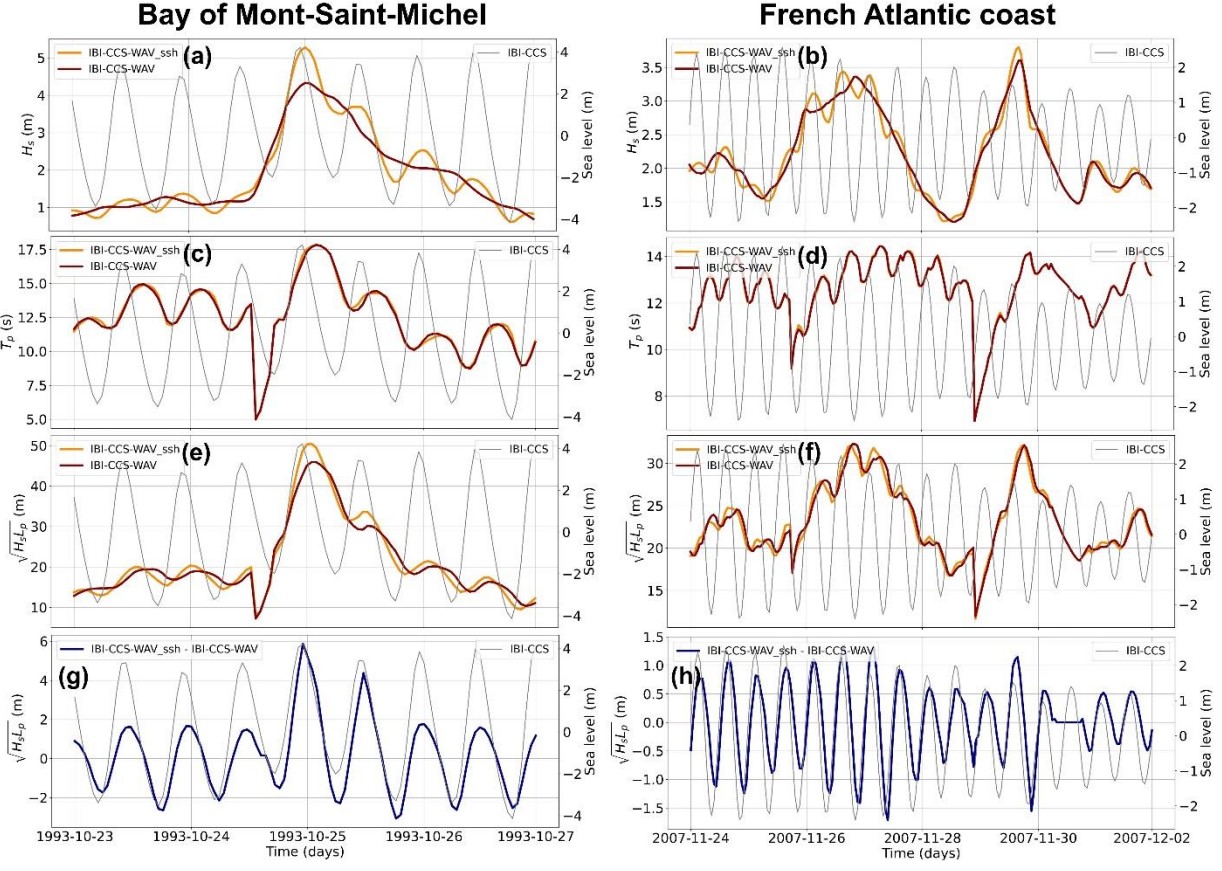





**Figure 11: Time series of incoming wave conditions for the Bay of Mont-Saint-Michel (first column) and for the French Atlantic coast (second column) during an extreme significant wave height event. The two locations are marked on Fig. 3b. The curves represent IBI-CCS-WAV (dark red curve) and IBI-CCS-WAV_ssh (dark yellow curve) and 3 variables are displayed: (a,b)**
**significant wave height (Hs, in m), (c,d) peak period (Tp, in s), (e,f) scaling for wave setup ($\sqrt{H_s L_p}$, in m) and (g,h) differences in the wave setup scaling between the two simulations. Water level variations are shown in thin gray lines, with the right y-axis on each panel.**

As shown in Figure 10, the most significant impact of including water level variations for wave modeling is found during extreme events. Figure 11 illustrates time-series of significant wave height, peak period, wave setup scaling and differences
in the wave setup scaling for two extreme significant wave height events at the same two macro-tidal locations. The events selected in the model did not actually occur since the GCM forcing is not in phase with the observed forcing. The significant wave height and wave setup scaling time-series from IBI-CCS-WAV_ssh oscillate in phase with the tide in IBI-CCS-WAV_ssh, illustrating the consideration of hourly water levels in the regional wave model (Fig. 11a,b,e,f). In the case of the Bay of Mont-Saint-Michel, due to the large tidal range, the highest significant wave height, reached on day 25/10/1993, is 1
m larger (+25 %) in IBI-CCS-WAV_ssh than in IBI-CCS-WAV. The impact of the water level variations on the peak period is however small (Fig. 11c). In both IBI-CCS-WAV and IBI-CCS-WAV_ssh, diurnal variations of the peak period appear due to tidal current that shortens or lengthens the dominant wave period (Ardhuin et al., 2012). The impact of the inclusion on the wave model of the hourly water level variations on the wave setup scaling (Fig. 11e) is balanced by the effects on the significant wave height and peak period of the water level variations. For the peak of day 25/10/1993, the differences of the
wave setup scaling between IBI-CCS-WAV and IBI-CCS-WAV_ssh can reach 6 m (Fig. 11g). Differences in the wave setup can reach +8.4 cm to +14.7 cm for the peak event using parameterization (1) with beach slopes of 4 % and 7 %. As the Bay of Mont Saint-Michel has one of the highest tidal ranges in the IBI domain, the impacts found correspond to the upper bound with the settings of our model. For the French Atlantic coast, due to a lower tidal range of 4 m, the impact of including the hourly water level variations on the extreme wave conditions is less important (Fig. 11, second column), as shown in Figure
10. For the event of 29/11/2007, differences between IBI-CCS-WAV_ssh and IBI-CCS-WAV are of +25 cm (+6%) for the significant wave height and of +1 m (+3 %) for the wave setup scaling (Fig. 11b,f). In terms of wave setup, this would result in differences between +1.4 cm and +2,45 cm with parameterization (1) which is substantially lower than in the bay of Mont-Saint-Michel.

**4.2 Impact of non-linear wave-water level interactions for the entire coastal domain over the full period**



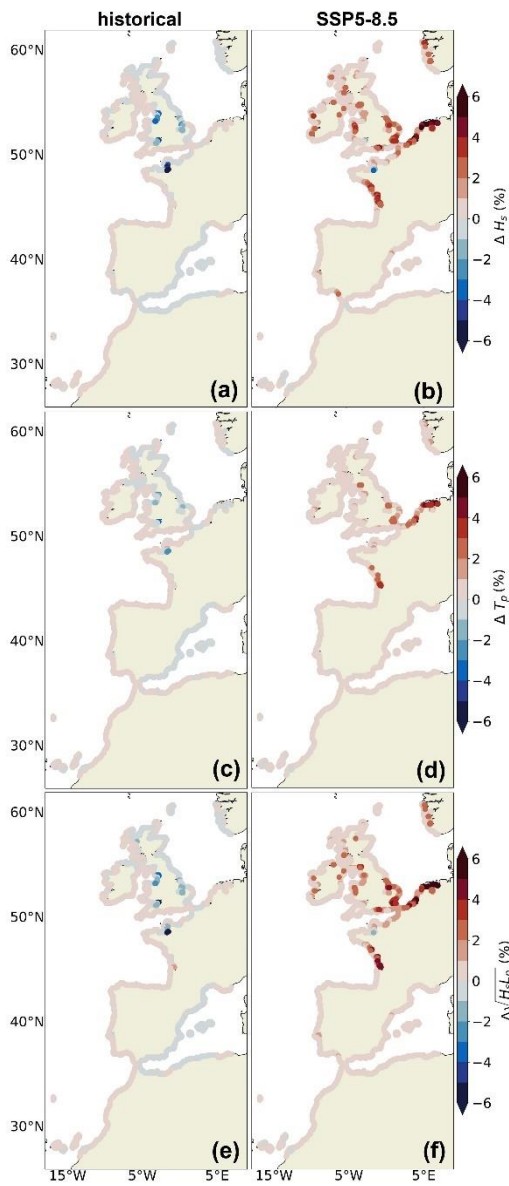

**Figure 12: Impact of the inclusion of the hourly water level variations in the wave model on the mean state of (a,b) significant wave height (first row, ΔHs, in %), (c,d) peak period (second row, ΔTp, in %) and (e,f) wave setup scaling (third row, $\Delta\sqrt{H_s L_p}$, in %). The first column shows the relative differences of mean state between IBI-CCS-WAV_ssh and IBI-CCS-WAV for the 1986-2005 period. The second column shows the relative differences of mean state between IBI-CCS-WAV_ssh and IBI-CCS-WAV for the 2081-2100 period under the SSP5-8.5 scenario.**

Figure 12 shows the impact of accounting for hourly water level variations in the wave model on the mean state of wave conditions by comparing IBI-CCS-WAV_ssh to IBI-CCS-WAV. Except for a few places, like the Mont Saint Michel or the




mouth of some rivers in the United Kingdom, the impact of including the water level variations on the wave model has no impact on average as shown in the first column of Fig. 12. This is due to the fact that the differences between IBI-CCS-WAV and IBI-CCS-WAV_ssh oscillate at the tide frequency around a zero-average value, as shown in Fig. 11. Furthermore, this suggests that there are no strong non-linear effects between waves and sea level that would make an impact on the mean state on average for the majority of the coastal domain.

Differences in the wave projections due to the inclusion of hourly water level changes are illustrated in the second column of Fig. 12 at the end of the century (2081-2100) under the SSP5-8.5 scenario. Sea level projected changes in the IBI-CCS ocean model during the 21$^{st}$ century are mainly dominated by the mean sea level rise, with rather small changes in tides and storm surges (Chaigneau et al. 2022). Therefore, as IBI-CCS-WAV and IBI-CCS-WAV_ssh are forced by the same winds, Figure 12 mostly shows the impact of mean sea level rise on waves, even if not completely linear. In IBI-CCS, the sea level rise averaged over the IBI domain reaches +80 cm in 2100 compared to the 1986-2005 mean under the SSP5-8.5 scenario considering changes in tides and storm surges too (Chaigneau et al., 2022). Until the end of the paper, the sea level rise term will also consider projected changes in the mean state of tides and storm surges. This long-term sea level rise has an overall effect on the coastal points of the large continental shelf where shallow-water dynamics prevail (Fig. 3a). The significant wave height projections are up to +3 cm larger (e.g. +6 % larger than in IBI-CCS-WAV) along the French Atlantic coasts and in the southern North Sea (Fig. 11b). This result agrees with Arns et al., 2017 who showed that water depth changes induced by sea level rise were leading to waves of greater amplitude and period, breaking closer to the shore. The impact of the sea level rise on the mean state of significant wave height leads to an impact of the same order of magnitude (up to +6 %) on the scaling wave setup mean state (Fig. 12f). For the peak period, the impact is still moderate (Fig. 12d). In the southern North Sea, projected changes in significant wave height are small (Fig 7b) and therefore an added impact of +3 cm due to the sea level rise corresponds to more than 70 % of the projected total changes under SSP5-8.5 scenario.



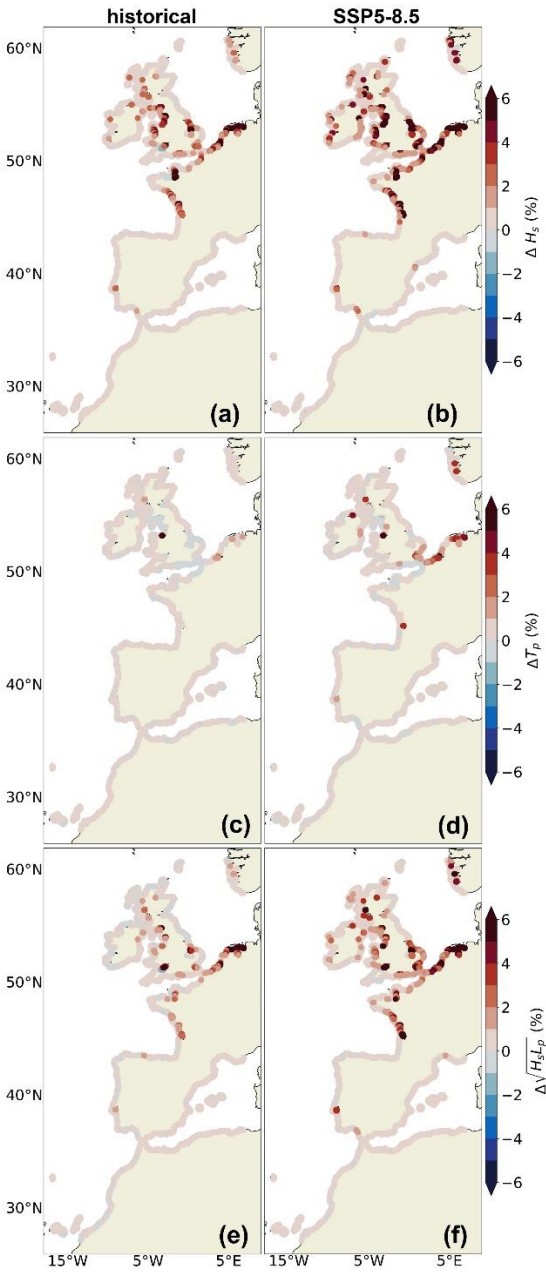

**Figure 13: Impact of the inclusion of the hourly water level variations in the wave model on the 99th percentile of (a,b) significant wave height (first row, ΔHs, in %), (c,d) peak period (second row, ΔTp, in %) and (e,f) wave setup scaling (third row, $\Delta\sqrt{H_s L_p}$, in %). The first column shows the relative differences of the 99th percentile between IBI-CCS-WAV_ssh and IBI-CCS-WAV for the 1986-2005 period. The second column shows the relative differences of the 99th percentile between IBI-CCS-WAV_ssh and IBI-CCS-WAV for the 2081-2100 period under the SSP5-8.5 scenario. Note that the color bars are saturated in red for some points for (a), (b) and (f).**



Figure 13 shows the impact of accounting for the hourly water level variations in the wave model on the 99[th] percentile by comparing IBI-CCS-WAV_ssh to IBI-CCS-WAV. The impact over the historical period is substantially more important when considering the 99[th] percentile instead of the mean state (Fig. 13, first column). The coastal points of the large continental shelf are highly impacted (southern North Sea, English Channel, seas around the United Kingdom, French

Atlantic coasts) and particularly macro-tidal locations (Fig. 3b) such as the Bay of Mont-Saint-Michel, the Bristol Channel and the eastern Irish Sea. In these areas, the historical 99[th] percentile of significant wave height and wave setup scaling is up to +8% and +6 % higher respectively when considering water level variations, mostly due to tidal variations.

The impact of including the hourly water level variations on the projections of the 99[th] percentile is even larger due to the combination of the impact of tidal range and sea level rise. For the significant wave height, the projected 99[th] percentile is

increased by up to +6 cm or +10 % and for the wave setup scaling it represents an increase of up to +8 % (Fig. 13 second column). As projected changes in the extreme significant wave heights and peak periods in IBI-CCS-WAV are quite small at the coast (Fig. 8b,c), especially on the French Atlantic coasts and on the North Sea coasts (<10 cm, <0.3 s), an impact of the hourly water level variations of +6 cm or +0.1 s represents more than 80 % of the projected changes in IBI-CCS-WAV_ssh.

### 4.3 Impact on extreme events in terms of return periods

It was shown in Sect. 4.2 that the impact of the inclusion of the hourly water level variations in the wave model had a larger effect on the 99[th] percentile than on the mean state of wave conditions. To better document the impacts on extreme events, we now focus on high return periods, such as the 100-year return level. In Sect. 4.3, a nonstationary extreme value analysis (EVA) is performed for each location time series by using the approach described by Mentaschi et al., 2016. This method is used to detect long-term trends in the extremes and to filter out the variability on time scales shorter than 30 years. For each

location and wave time series, the output of the EVA analysis is a time-varying generalized Pareto distribution (GPD).



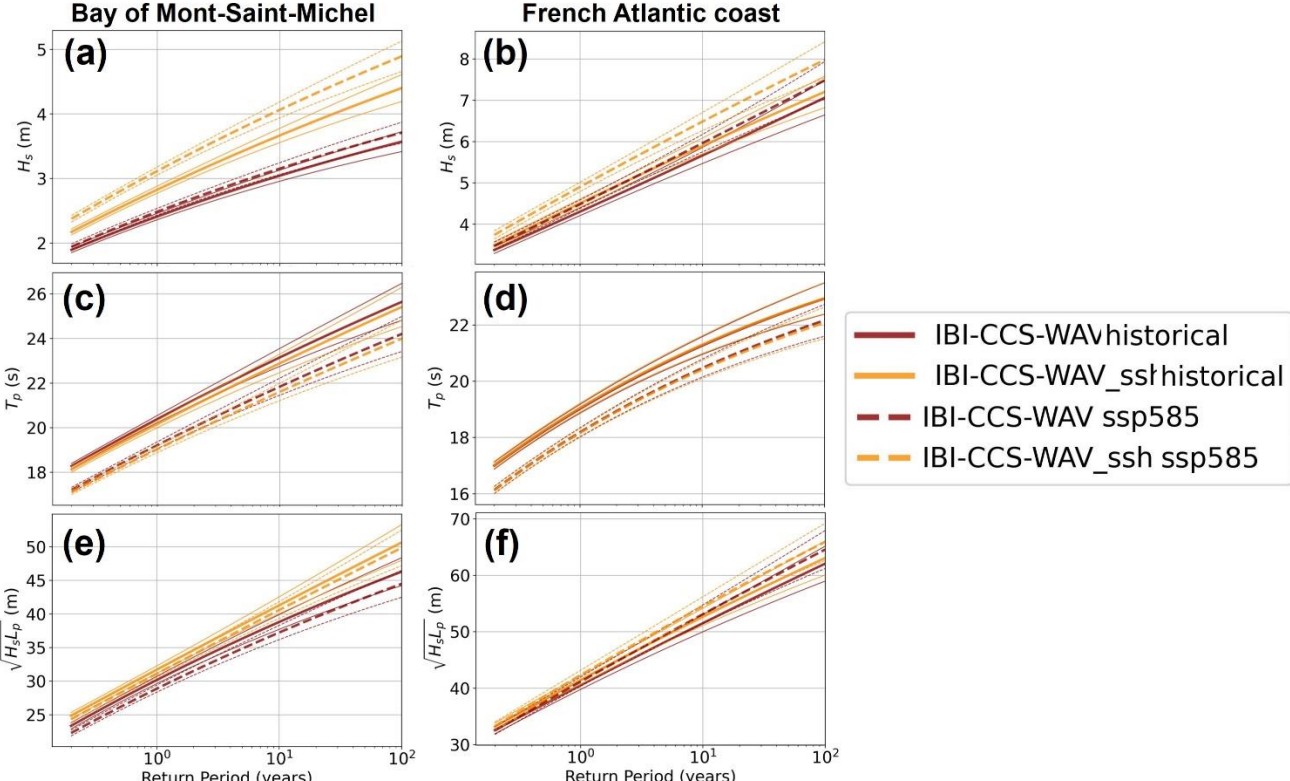

**Figure 14: Return period curves of incoming wave conditions for IBI-CCS-WAV (dark red curves) and IBI-CCS-WAV_ssh (dark yellow curves) for the Bay of Mont-Saint-Michel (first column) and for the French Atlantic coast (second column) (Fig. 3b). The solid lines represent the year 1985 (representative of the 1970-2000 period) and the dashed lines the year 2085 (representative of the 2070-2100 period) for the SSP5-8.5 scenario. The rows show 3 variables: (a,b) significant wave height (Hs, in m), (c,d) peak period (Tp, in s) and (e,f) scaling for wave setup ($\sqrt{H_s L_p}$, in m). The thin solid and dashed lines are the confidence intervals (corresponding to 1 sigma confidence) associated with the extreme value analysis (EVA).**

Figure 14 shows the return period curves for the Bay of Mont-Saint-Michel and for the French Atlantic coast for the two simulations IBI-CCS-WAV and IBI-CCS-WAV_ssh. In the Bay of Mont-Saint-Michel, the differences between the two simulations are very important for the 1970-2000 period (dark curves) for the significant wave height (Fig. 14a). It is especially the case for high return periods such as the 100-year return level. As the confidence intervals for the two simulations are disjoint, the differences are considered significant. For the 2070-2100 period under the SSP5-8.5 scenario, the differences between the two simulations are even larger due to the sea level rise of about +80 cm. For example, in the case of the significant wave height, the 100-year return level is +35 % larger when considering the water level variations and for the wave setup scaling it represents an increase of +11 %. In terms of wave setup, using parameterization (1), the differences in the wave setup can reach from +7 cm to +12.25 cm for the 100-year return level depending on the beach slope. This value seems small but is important to consider in the context of threshold exceedance calculations to predict coastal flooding. For the French Atlantic coast, the confidence intervals between IBI-CCS-WAV and IBI-CCS-WAV_ssh are not



distinct. Therefore, the differences due to the inclusion of the water level variations on the wave model are not considered
significant. Overall, the longer the return periods, the larger the differences between the two simulations.

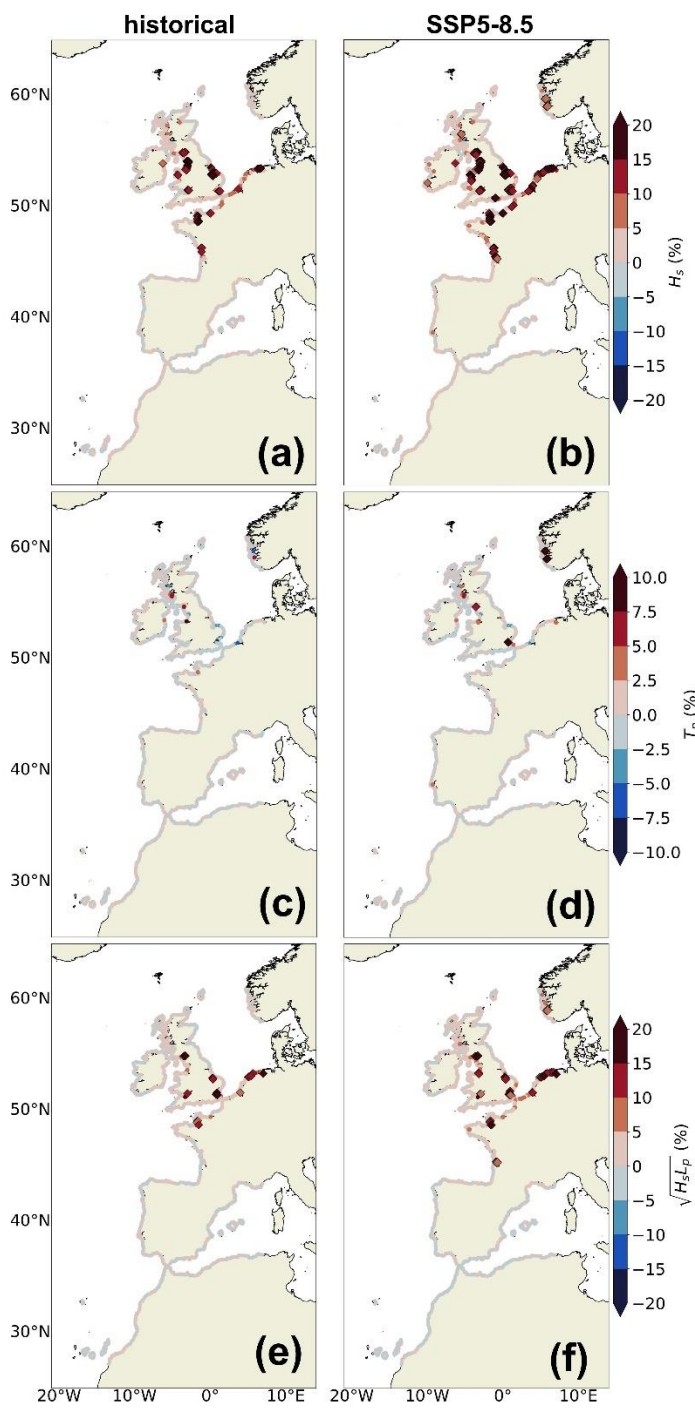





**Figure 15: Impact of the inclusion of the hourly water level variations in the wave model on the 100-year return level of (a,b) significant wave height (first row, ΔHs, in %), (c,d) peak period (second row, ΔTp, in %) and (e,f) wave setup scaling (third row, $\Delta\sqrt{H_s L_p}$, in %). The first column shows the relative differences of the 100-year return level between IBI-CCS-WAV_ssh and IBI-**
**CCS-WAV for the year 1985 (representative of the 1970-2000 period). The second column shows the relative differences of the 100-year return level between IBI-CCS-WAV_ssh and IBI-CCS-WAV for the year 2085 (representative of the 2070-2100 period) under the SSP5-8.5 scenario. The large diamonds represent the locations where the differences between both simulations are significant (i.e. where the confidence intervals associated with the 100-year return level calculation are disjoint). Note that the colorbars are saturated in red for some points for (b).**

Projected changes in the 100-year return level under SSP5-8.5 scenario (not shown) are globally consistent with projected changes in the 99[th] percentile of significant wave height, peak period and wave setup scaling of Figure 8. Therefore, we use the IBI-CCS-WAV and IBI-CCS-WAV_ssh simulations to assess the influence of the inclusion of hourly water level variations in the wave model during high return period events. As for the 99[th] percentile of Figure 13, the coastal points of the large continental shelf are highly impacted and particularly macro-tidal locations (Fig. 3b). However, the differences

between IBI-CCS-WAV and IBI-CCS-WAV_ssh for the 100-year return level are of a larger amplitude than the differences for the 99[th] percentile of Figure 13 for the whole domain. At the end of the century, the consideration in the wave model of the combination of the tidal range, storm surges and sea level rise lead to greater values in extreme significant wave height and wave setup scaling by up to +20 % and +10 % respectively. However, these large impacts are found in locations where the projected changes for the 21[st] century are generally small since the largest projected changes are located in the southern

domain. In spite of this, the effect of sea level on waves should be important to consider when analyzing extreme wave events but also when analyzing EWL events even if fewer locations are concerned for the wave setup scaling.

## 5 Discussion

### 5.1 Model resolution

In our study, the impact of including the hourly water level variations in the wave model is limited by several modeling
aspects. The first limitation is the horizontal resolution of the wave model. The model resolution of 1/10 ° does not allow a very fine representation of the bathymetry in the coastal zone and of the coastline. For instance, it would be unrealistic to have a bathymetry of 1 m within a 10km grid point. In consequence, a wave model with a horizontal resolution of 1/10 ° has fewer shallow water areas than a higher resolution model. In addition, the model resolution here does not allow the modeling of wave transformations in the coastal zones which can affect the wave projected changes at the coast.

### 5.2 Model dry areas

Another limitation in the modeling framework is that the regional ocean model IBI-CCS does not allow for dry areas. Therefore, a minimal bathymetry is set to 6 m to run the ocean model with tides (Chaigneau et al. 2022). We chose to apply the same minimal bathymetry of 6 m in the regional wave model to maintain consistency between both regional ocean and wave models. This results in fewer shallow water areas in the wave model which also limits the effect of the water level





variations on waves. The implementation of "wetting and drying" (O'Dea et al., 2020) allowing for dry areas in NEMO version 4.2 should improve this limitation.

### 5.3 Impact of waves on sea level

The aim of the study was to better understand the non-linear interactions between waves and sea level. In the modeling framework of the paper, only the effect of sea level on waves is accounted for. However, both are coupled in reality and

waves also have a feedback on sea level. For instance, Bonaduce et al., 2020 have studied in the European Seas the contribution of wave processes to sea level variability with ocean-wave coupled simulations at an eddy-resolving spatial resolution of 3.5 km. They highlighted the occurrence of mesoscale features of the ocean circulation and a modulation of the surge at the shelf break due to the effect of the wave forcing on sea level. More importantly, they also reported a large contribution of wave induced processes to sea level extremes.

**5.4 Storm wave conditions with atmospheric fields from a GCM**

In this study, we have investigated projected changes in wave extreme events (99th percentile and 100-year return level). However, the typical CMIP5 and 6 (Coupled Model Intercomparison Project fifth and sixth phases) climate model atmospheric resolutions (mostly 1 or 2 °) do not allow the generation of intense tropical/extratropical storms responsible for severe storm wave conditions (Seneviratne et al., 2021; Morim et al., 2019). In our case, the use of a GCM with a higher

spatial resolution compared to the typical coarse resolutions of CMIP5 and 6 models was interesting for both the ocean (1/4 °, eddy permitting) and atmosphere (1/2 °). In particular, the atmospheric forcing applied to the regional wave model should increase the realism of the forcing compared to other GCMs. For instance, the intensity of the atmospheric low-pressure systems and the spatial patterns generating extreme wave episodes should be better reproduced.

### 5.5 Use of a single forcing GCM

However, the use of a single forcing GCM and a single member does not allow to quantify the uncertainties of the projected changes. Here, the aim of the study was not to characterize the uncertainties nor provide a likely range of wave projected changes over the IBI domain. Rather, the regional model was used to investigate questions related to sea level processes in a changing climate. To gain insight into the representativeness of the GCM forcing model chosen here, we verified that the wave projected changes (significant wave height and peak period) were in good agreement with other studies over the IBI

domain. We also checked at different locations that the projections of extreme winds in CNRM-CM6-1-HR were consistent with projected changes of other CMIP6 models (Sect. 2.2.2). Furthermore, the differences in the regional wave projections with and without the inclusion of the water level variations provide an indication of another source of uncertainty of the modeling chain when analyzing regional climate simulations.





## 6 Conclusions

Several studies have shown that water depth changes induced by sea level rise can induce changes in the wave field (Idier et al., 2019; Arns et al., 2017; Hoeke et al., 2015). The aim of the present paper was to investigate the impact of including hourly water level variations (tides, storm surge, mean sea level rise) on wind-waves characteristics and wave setup in a regional wave model of the northeastern Atlantic over the 1950-2100 period. As a first step, the regional wave model has been presented and validated over the 1993-2014 period. Comparisons to observations and a wave reanalysis showed an

overall good performance of the model. Secondly, projected regional changes in mean and extreme wave conditions were presented. They were shown to be consistent with previous studies, with a general decrease in mean and extreme significant wave height and peak period over the northeastern Atlantic region. Finally, the impact of including hourly water level variations in the wave model was assessed over the historical period and for 21[st] century projections for the mean state and extremes of wind-wave characteristics and wave setup, the latter contributing to coastal sea level hazard such as coastal

flooding.

For the historical period, wave-water level interactions on the mean wave state and wave setup were found to be very small except for a few locations. The impact of water level changes on wind waves over the historical period is substantially more important when considering the 99[th] percentile or 100-year return level along the coasts of the large continental shelf and particularly in large tidal range areas. For example, in the Bay of Mont-Saint-Michel where the tidal range is of 10 meters,

extreme significant wave heights were found to be larger by 1 meter (or +25 %) during a historical extreme event when considering hourly water level variations in the wave model. The corresponding increase in wave setup reached +8.4 cm and +14.7 cm, when considering beach slopes of 4 % and 7 % respectively. However, these values are the upper bound of the sensitivity of significant wave height and wave setup to wave-water level interactions with the settings of our model, as the Bay of Mont-Saint-Michel is subject to one of the largest tidal ranges of the IBI domain.

Mean sea level rise (and changes in tidal amplitudes and storm surges) reaches +80 cm in 2100 compared to the 1986-2005 mean under the SSP5-8.5 high-emission climate change scenario over the northeastern Atlantic (Chaigneau et al. 2022). This mean sea level rise has an overall effect in our regional projections of the mean wave state over the large continental shelf where shallow-water dynamics prevail. Significant wave heights are projected to be up to +3 cm (or +6 %) higher along the French Atlantic coasts and in the southern North Sea by the end of the 21[st] century due to water depth changes. Sea level rise

indeed relaxes depth-induced breaking of wind-waves and leads to waves of greater amplitude breaking closer to the shore. The impact of accounting for wave-water level interactions on the scaling wave setup mean state is of the same order of magnitude (+6 %). However, it should be noted that these impacts are found in locations where projected changes in significant wave height over the 21st century are generally small since the largest projected changes are located in the southern part of the regional domain. The major impact of the inclusion of hourly water level variations in the wave model is

found during extreme events (99[th] percentile or 100-year return level) at the end of the 21[st] century. For high return period



events, the consideration in the wave model of the combination of tides, storm surges and sea level rise lead to greater values in extreme significant wave height by up to +20 % for many locations at the end of the 21$^{st}$ century. The impact on the wave setup scaling can be up to +10 % but fewer locations are concerned. Overall, the inclusion of water level variations on the wave model had almost no impact on the peak period. However, the study is subject to limitations mainly in terms of model

horizontal resolution (1/10 °) and associated coarse bathymetry which limit the non-linear interactions between waves and water level changes whether from tides, storm surges, or long-term sea level rise.

In conclusion, our results advocate for the inclusion of wave-water level non-linear interactions in modelling studies of wave extremes, in particular when significant wave heights are of interest. These non-linear interactions should be accounted for when threshold exceedances are calculated for example in order to prevent coastal flooding or to build coastal protection

structures in a climate change context.

**Code availability**

The MFWAM model used in this study is based on the wave model WAM freely available at https://github.com/mywave/WAM.

**Data availability**

Information on CNRM-CM6- 1-HR simulations can be found at https://doi.org/10.22033/ESGF/CMIP6.4067 (CNRM-CM6-1-HR, historical; Voldoire, 2019a), https://doi.org/10.22033/ESGF/CMIP6.4164 (CNRM-CM6-1-HR, piControl; Voldoire, 2019b), https://doi.org/10.22033/ESGF/CMIP6.4185 (CNRM-CM6-1-HR, ssp126; Voldoire, 2020a), https://doi.org/10.22033/ESGF/CMIP6.4225 (CNRM-CM6-1-HR, ssp585; Voldoire, 2019c). The CNRM-CM6-1-HR forcing fields are available on the ESGF website (ESGF, 2022a: historical data, http://esgf-data.dkrz.de/search/cmip6-

dkrz/?mip_era=CMIP6&activity_id=CMIP&institution_id=CNRM-CERFACS&source_id=CNRM-CM6-1-HR&experiment_id=historical; ESGF, 2022b: piControl data, http://esgf-data.dkrz.de/search/cmip6-dkrz/?mip_era=CMIP6&activity_id=CMIP&institution_id=CNRM-CERFACS&source_id=CNRM-CM6-1-HR&experiment_id=piControl; ESGF, 2022c: ssp126 data, http://esgf-data.dkrz.de/search/cmip6-dkrz/?mip_era=CMIP6&activity_id=ScenarioMIP&institution_id=CNRM-CERFACS&source_id=CNRM-CM6-1-HR&experiment_id=ssp126;

ESGF, 2022d: ssp585 data, http://esgf-data.dkrz.de/search/cmip6-dkrz/?mip_era=CMIP6&activity_id=ScenarioMIP&institution_id=CNRM-CERFACS&source_id=CNRM-CM6-1-HR&experiment_ id=ssp585). The reanalysis data and wave buoy observations were obtained from the Copernicus Marine Services (Copernicus, 2022a: reanalysis data, https://doi.org/10.48670/moi-00030; Copernicus, 2022b: observational data, https://doi.org/10.13155/53381).



**Author contribution**:

AM designed the study. LA prepared the regional wave model configuration. SLC adapted the regional wave model to consider hourly variations of sea level and performed the regional wave simulations. AAC performed the sea level regional simulations and did the analyses of the study. AM, AV, GR, SLC and LA supervised the project. AM wrote the introduction, SLC wrote the Methods section and AAC wrote the Results, Discussion and Conclusions sections. All authors contributed to
manuscript revisions and read and approved the submitted version.

**Competing interests**:

All authors declare that they have no conflicts of interest.

**Acknowledgements**

Analyses were carried out with Python. The authors thank Joanna Staneva for her advice and help on the implementation of
the sea level forcing in the regional wave model.

**Financial support**

The PhD thesis of AAC is supported by Mercator Ocean and Météo-France.

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
