# Peer review of "Impact of wave-water level non-linear interactions for the projections of mean and extreme wave conditions along the coasts of western Europe"

_EGUsphere, 2022_

## Author Comment (AC1)

20.12.2022

Answer RC1:

The authors wish to thank the anonymous reviewer for his/her thorough assessment and constructive comments which greatly helped us to improve the quality of the paper. We are pleased to address the point-by-point answers to your review in blue in the supplement to this comment.

Best regards,

The authors.

**Main comments:**

1. Methodology:

- As stated by the authors (L60-61), "wave characteristics used to estimate wave setup are sensitive to water level changes in shallow waters, where waves interact with the ocean bottom." From section 2.2.4 I understand that the wave model considers water-level variations only for the wave propagation (i.e., group velocity and wave number) while "coastal (depth-induced) breaking is not included" in the model (L92-95). My concern is that by not including the depth-induced wave-breaking the authors are missing a fundamental depth-dependent process, which can have a first order effect on the wave statistics in shallow water and hence on the wave setup. In addition, as the authors also explain in the introduction (L35-36), the wave setup is in fact due to the depth-induced wave breaking. So what I cannot understand is how the authors can assess the impact of water-level variations on the wave setup if the leading order physical mechanism driving the wave setup is not included in the model. I think the authors should carefully address this point in their manuscript.

The Introduction, Sect. 2.1 (Description of the wave model) and Discussion have been revised to clarify this point by (1) highlighting more clearly the processes taken into account in the model that are likely to be impacted by non-linear interactions and those that are not activated (2) modifying the introduction which focused too much on the coastal wave breaking compared to the model's capabilities and the purpose of the study.

To explain the choice not to activate the depth-induced wave breaking in the model, the two following sections have been added in the Discussion:

"**5.3 Limitations associated with the applicability of the parametrization to coastal points**

To calculate the wave setup for the coastal points of our regional domain, we chose to use the simple parameterization of Stockdon et al., 2006 based on deep water parameters. However, the coastal points are theoretically not purely deep water as shown in Figure 2a (yellow dotted lines). Yet, this approach, used in other climate studies (Melet et al., 2018, 2020a; Lambert et al., 2020), appears pragmatic given the model resolution limitations (Sect. 5.1) and the processes accounted for in the wave model (Sect. 2.1).

**5.4 Impact of the absence of depth-induced wave breaking**

Very close to the coast, the depth-induced wave breaking is a fundamental depth-dependent process that can have a first-order effect on the shallow water wave statistics and thus on the wave setup. As explained in Sect. 2.1, the physics associated with the explicit representation of coastal breaking waves is not activated. Such an approach is justified because our primary interest is to calculate the wave setup contribution to include it in further analyses on extreme water levels and the parameterization used to compute this contribution is based on deep water parameters (Sect. 2.5) that are not supposed to be affected by coastal wave breaking.

With the coastal wave breaking included, the significant wave heights should be substantially impacted in shallow waters. The impact of the inclusion of the water level variations in the wave model would also probably differ. A perspective for this study would be to take into account coastal wave breaking and to apply new specific wave setup formulations which would not require deep water characteristics or to use a wave model that directly resolve the wave setup."

Some tests have been performed with the coastal wave breaking activated (Battjes and Janssen, 1978) in the regional wave model for the significant wave extreme event of the year 1993 in the Bay of Mont Saint Michel (Fig. RC1). These tests suggest that the conclusions of larger extreme significant wave heights due to the inclusion of the water level variations occurring at high tide are still qualitatively valid. Nevertheless, the impact is larger with the coastal wave breaking activated, which is expected to be even more significant at the end of the century with the mean sea level rise.

[Figure]

**Figure RC1: Same as Fig. 12a of the paper but with new simulations that include the depth-induced coastal wave breaking: IBI-CCS-WAV_ssh_cwb and IBI-CCS-WAV_cwb.**

- When assessing the impact of water-level forcing on the wave setup at a domain level the authors report an impact in few coastal locations. My doubt is: how much can we trust these results given the 6m minimum depth approximation?

Sect. 5.1 has been revised:

**"5.1 Model resolution limitations**

In our study, the impact of including the hourly water level variations in the wave model is limited by several resolution aspects. The first limitation is the horizontal resolution of the wave model. The model resolution of 1/10 ° (~10 km) is conditioned by the computational cost due to the length of the simulations needed to address the question of extremes in a climate scale. It does not allow a very fine representation of the coastline and of the bathymetry in the coastal zones. For instance, to maintain a realistic balance between the 10 km horizontal resolution and the water depth, the minimum bathymetry is set to 6 m (i.e. time-mean minimum of 6 m) because it would have been unrealistic to have a bathymetry of 1 m within a 10 km grid point. In consequence, the wave model has fewer intermediate and shallow water areas than a higher resolution model and thus less non-linear interactions. […] Therefore, the results are not representative of the real situation at the coast but rather give a regional information."

- Also, could it be that the authors found a generally small (very few locations) impact because the depth-induced breaking is neglected and the minimum depth approximation is applied?

Our work highlights that wave-water level non-linear interactions exist and should be considered for applications on extreme wave events for large tidal range areas and especially for climate applications due to future sea level rise. However, as stated by the reviewer, due to the resolution of the model and therefore the limit on the minimal bathymetry, the estimates provided represent only a part of the processes responsible for the wave-water level non-linear interactions.

A sentence has been added in Sect. 5.4:

"As the regional wave model does not have a very fine representation of the bathymetry or the coastline (Sect. 5.1) and does not resolve the wave transformations in the coastal zones, the estimates provided in this study only partially represent the processes responsible for the wave-water level non-linear interactions."

2. Validation:

- The title of section 3 is "Validation and projections of IBI-CCS-WAV, without wave-water level interactions" and in fact the figures of this section report data only for IBI-CCS-WAV. However, at L259-261 the authors state that "The ability of IBI-CCS-WAV and IBI-CCS-WAV_ssh to reproduce observed distributions is assessed for the mean state and the 99th percentile of the significant wave height and peak period since these variables are then used to compute the wave setup scaling". Is the IBI-CCS-WAV_ssh validated as well? If not (as I believe is the case), then I think the authors should also include the validation for the experiment using water-level forcing since, apart from the impact on the wave setup, it is also interesting and useful to know for the wave modelling community whether including this forcing can help to improve the accuracy of the model.

The validation of the IBI-CCS-WAV_ssh simulation has been added only on the two scatter plots of Figure 4 and 5 which are the comparisons to tide gauge data.

A paragraph has been added in Sect. 3.1.1 : "The comparison of IBI-CCS-WAV_ssh with the reanalysis is not relevant since the latter does not consider the forcing with hourly sea level variations. The IBI-CCS-WAV_ssh simulation is compared to the buoy data in Figure 4d,h and Figure 5,d,h. However, it is difficult to get useful information from these comparisons with buoys since they are not located at the coast. Actually, they are in areas where there is no impact of the wave-water level non-linear interactions (Sect. 4) so the performances of IBI-CCS-WAV_ssh are similar to those of IBI-CCS-WAV."

For the same reasons and also because we found that the inclusion of the water level variations on the wave model has no impact on the mean wave direction over the 1993-2014 period, we chose to not include the wave rose of IBI-CCS-WAV_ssh. A sentence is added Sect. 3.1.2 : "In Figure 6, the focus is only on the IBI-CCS-WAV simulation as we found that the impact of the water level variations on the mean wave direction was negligible over the 1993-2014 period."

3. Manuscript structure:

Thank you very much for this comment which has significantly improved the structure of the manuscript.

- In general, I think the structure of the paper should be substantially improved before being suitable for publication. Below, a list of possible changes:
- Section 2: this section is quite confused and not logically structured in my opinion. I would first move L101-112 as in intro of Sec. 2, improving the text and Fig 2 (the colours are to weak). Then, I think the authors could

a) describe the numerical wave model (sec 2.1), avoiding the references to global and regional simulations (e.g. L85), since I think can confuse the reader. Done.

b) describe the regional wave configuration IBI-CCS-WAV (sec 2.2): this is the real focus of this paper, all the other models are used to force this model in my opinion. Done. In addition, I would move L185-190 at the beginning of this section just to state at the beginning what is the aim of this model.

The authors finally decided to first describe the zone, the processes involved, and then to explain the downscaling methodology.

c) describe the external forcings (sec 2.3) with three subsections:

*) Atmospheric forcing (sec. 2.3.1), describing and validating (L138-152) CNRM-CM6-1-HR model and the fields used to force IBI-CCS-WAV. Also, please avoid the acronym GCM which is typically used for General Circulation Model instead.

Done. The acronym GCM has been removed.

*) Hydrodynamic forcing (sec 2.3.2), describing IBI-CCS and the fields used to force IBI-CCS-WAV. Done.

*) Wave forcing (sec 2.3.3), describing CNRM-HR-WAV and the fields used to force IBI-CCS-WAV.
Done.

d) Inclusion of water level variations in the regional wave model: IBI-CCS-WAV_ssh (sec 2.4). Done.

e) Wave setup calculation (sec. 2.5): Please check the definition of the wave setup scaling – there is a delta in the definition (L243) that I think should not be there.

The delta has been removed.

- Section 4: I would first describe the impact on the entire coastal domain and after on the specific locations. Also, I think the authors should clarify better what is the rational behind the choice of those two specific locations. Why not for example the Bristol channel? The tidal range there is almost as large as in Mont-Saint Michel. Also, I would rewrite Sec. 4.2 and 4.1 (which are the most important sections in my opinion), trying to discuss more in depth what is the impact and to contextualise it, maybe moderating a bit the wording (e.g., "highly impacted") which I think it is not fully reflecting the results of the authors.

The impact on the entire domain is provided before the specific locations. The two specific locations are chosen in France as it is our country of interest here but in terms of processes, indeed, the Bristol Channel would also have been relevant for the effects of the large tidal range. The wording has been moderated in Sect. 4.1 and 4.2.

A paragraph has been added in Sect. 4.2

"However, it can be pointed out that the most significant increase in wave height occurs in both cases at high tide. These results are in agreement with Lewis et al., 2019 and Calvino et al., 2022 who both showed a significant increase in wave height at high tide at a finer scale. In Calvino et al., 2022, this impact seems to be explained mainly by the effect of bottom friction, which is less important at high tide as there is more water. In our case, additional analyses would be needed to understand which is the primary process included in the model (Sect. 2.1) responsible for the non-linear interactions."

Also, another section has been added in the Discussion to discuss more deeply the results:

"**5.6 Implications of the results on extreme wave projections**

Marine flooding hazards cannot be quantified based on wave setup alone but wave setup can locally partially balance or enhance water levels at the coast (Melet et al., 2020). Depending on the location (wave regimes, local ocean processes involved, sign of the extreme wave projected changes, amplitude of the projected changes in ocean processes), the inclusion of the non-linear interactions could thus enhance or balance the future wave extremes and may be important to consider for future flooding hazard calculations. The results presented in this study highlight that wave-water level non-linear interactions can be substantial for extreme wave height and wave setup, but are region dependent. For instance, the extreme wave projections are directly dependent on the water level variations forcing. In our case, the future water level variations and therefore a large part of the non-linear interactions are mainly associated with the mean sea level rise of about +80 cm and less so to changes in tides or storm surges. In other regions, large projected changes in tides or storm surges could impact the future waves conditions. For instance, Pickering et al., 2017 and Haigh et al., 2019 showed changes up to + 20 cm in the M2 component in the China Sea and in the Gulf of Saint Lawrence. Then, the future wave extremes could also be substantially more impacted in areas subject to larger mean sea level rise such as along the eastern coasts of the United States, in the Gulf of Mexico and in the Caribbean Sea where a rise of +1.4 m is expected at the end of the century under scenario SSP5-8.5 (Fox-Kemper et al., 2021)."

I would describe a bit better in the Conclusions and Abstract the limitations of your study.

A sentence has been added at the end of the Abstract: "However, as the wave setup is computed with a parameterization based on offshore characteristics, the depth-induced wave breaking is not activated in the model. The estimates provided in this study therefore only partially represent the processes responsible for the wave-water level non-linear interactions."

The end of the Conclusion has been revised: "However, as the regional wave model does not have a very fine representation of the bathymetry or the coastline and does not resolve the depth-induced wave breaking in the shallow areas, the estimates provided in this study only partially represent the processes responsible for the wave-water level non-linear interactions. Moreover, the results found might be dependent on the parametrization used to compute the wave setup and therefore on the beach slopes.

**Specific comments**

- L13: you don't need the acronym EWL here, since you don't use it anymore in the abstract.

Done.

- The authors may want to add some references at L170 – 173. Here I am suggesting some possible references for the North Atlantic (which is the area I am more familiar with): the Atlantic coasts are subject to very energetic events in terms of significant wave heights, wave periods and energy flows (e.g., Masselink et al. 2016, Bruciaferri et al. 2021) whereas the Mediterranean Sea and North Sea are more sheltered areas. In addition, the zone also contains very different tidal regimes with both macro and micro tidal regimes respectively in the English Channel/Celtic Sea (Valiente et al. 2018, Stokes et al. 2021) and in the Mediterranean Sea.

Some references have been added : "the Atlantic coasts are subject to very energetic events in terms of significant wave heights, wave periods and energy flows (Masselink et al., 2016; Bruciaferri et al., 2021) whereas the Mediterranean Sea and North Sea are more sheltered areas dominated by wind waves (Chen et al., 2002; Bergsma et al., 2022). In addition, the zone also contains very different tidal regimes with both macro and micro tidal regimes respectively in the English Channel/Celtic Sea (Valiente et al., 2019; Stokes et al., 2021) and in the Mediterranean Sea."

- L219-220: "Limitations related to the use of parameterizations have been extensively discussed in Melet et al., 2020" -> can the authors do a summary of those limitations here so that the reader is aware?

The limitations associated with the use of parameterizations (L229-237) have been moved to the Discussion section and completed by:

"**5.2 Limitations associated with the use of parametrizations for the wave setup**

Limitations in the use of parametrizations to estimate wave setup are thoroughly discussed in Melet et al., 2020; Lambert et al., 2020, including sensitivity analyses of the wave setup and runup contributions to different empirical parametrizations. The generic parametrization of Stockdon et al., 2006 used to compute the wave setup in our study is indeed subject to intrinsic limitations. A major limitation is that the formulation is only representative of sandy beaches. Other parameterizations (Guza and Thornton, 1981; Holman, 1986; dissipative case of Stockdon et al., 2006 or for a review Dodet et al. 2019) exist but they are often limited to specific coastal environments (e.g. dissipative sandy beaches, rocky cliffs) and have been calibrated with relatively few field data. The calibration therefore does not cover all the spectra of the environmental conditions.

For our large-scale study, another major limitation is that the parameterization relies on the specification of a beach slope. [L229-237]".

- L224: please define "foreshore".

The sentence has been modified: "where $\beta$ is the foreshore beach slope (i.e. the slope in the swash zone)".

- L355: Figure 7 illustrates "projected **changes**" -> changes respect to what? Please clarify

"Projected changes in incoming waves conditions from 1986-2005 to 2081-2100" has been replaced by "Projected changes in incoming waves conditions for the 2081–2100 period (relative to 1986–2005)." Same for Figures 8 and 9.

- L422-425: please rephrase it.

L422-425 have been rephrased: "In Sect. 3.2, as reported in other studies, we observed a general decrease in mean and extreme significant wave height and peak period over the domain and a clockwise mean wave direction change along the French Atlantic coasts. As IBI-CCS-WAV seems to be representative of other studies, we can used it to assess methodological modelling questions such as the impact of considering the hourly water level variations on the wave model."

- L453: the most significant impact -> quite strong wording, you have an impact (not so strong) only in one location out of two in Fig. 10.

"the most significant impact" has been replaced by "the larger impact".

- L461: however small -> to me seems nihil

"is however small" has been replaced by "is however almost zero".

- L484-485: I would be careful here. If what the authors are saying is true, then why it is not valid everywhere, e.g. Mont-Saint Michel? Please clarify.

The sentence has been removed and we need to further investigate the impact on the mean state, maybe with a high tide/low tide analysis to assess if there is an asymmetry of the impact of the water level variations on waves.

**References**

[revised manuscript text omitted]

---

## Author Comment (AC2)

20.12.2022

Answer RC2:

The authors wish to thank the anonymous reviewer for his/her review. The comments and remarks helped us improving the quality of the paper. We are pleased to address the point-by-point answers to your review in blue in the supplement to this comment.

Best regards,

The authors.

**Main comments:**

I thinks this paper brings an interesting contribution to the field, but before publication I would like the authors to clarify the following major point:

- The model used does not include shallow water processes such as wave breaking (line. 93), and cannot represent important interactions with the seabed in shallow regions, as the minimum depth is 6m (line. 206). These shallow water processes, as stated by the authors in the introduction, are important for wave setup and set down; yet this work estimates the setup from data that exclude them. Can the estimation of the setup, calculated excluding important shallow water processes, be trusted? Is it a reliable approximation?

The Introduction, Sect. 2.1 (Description of the wave model) and Discussion have been revised to clarify this point by (1) highlighting more clearly the processes taken into account in the model that are likely to be impacted by non-linear interactions and those that are not activated (2) modifying the introduction which focused too much on the coastal wave breaking compared to the model's capabilities and the purpose of the study.

To explain the choice not to activate the depth-induced wave breaking in the model, the two following sections have been added in the Discussion:

"**5.3 Limitations associated with the applicability of the parametrization to coastal points**

To calculate the wave setup for the coastal points of our regional domain, we chose to use the simple parameterization of Stockdon et al., 2006 based on deep water parameters. However, the coastal points are theoretically not purely deep water as shown in Figure 2a (yellow dotted lines). Yet, this approach, used in other climate studies (Melet et al., 2018, 2020a; Lambert et al., 2020), appears pragmatic given the model resolution limitations (Sect. 5.1) and the processes accounted for in the wave model (Sect. 2.1).

**5.4 Impact of the absence of depth-induced wave breaking**

Very close to the coast, the depth-induced wave breaking is a fundamental depth-dependent process that can have a first-order effect on the shallow water wave statistics and thus on the wave setup. As explained in Sect. 2.1, the physics associated with the explicit representation of coastal breaking waves is not activated. Such an approach is justified because our primary interest is to calculate the wave setup contribution to include it in further analyses on extreme water levels and the parameterization used to compute this contribution is based on deep water parameters (Sect. 2.5) that are not supposed to be affected by coastal wave breaking.

With the coastal wave breaking included, the significant wave heights should be substantially impacted in shallow waters. The impact of the inclusion of the water level variations in the wave model would also probably differ. A perspective for this study would be to take into account coastal wave breaking and to apply new specific wave setup formulations which would not require deep water characteristics or to use a wave model that directly resolve the wave setup."

Some tests have been performed with the coastal wave breaking activated (Battjes and Janssen, 1978) in the regional wave model for the significant wave extreme event of the year 1993 in the Bay of Mont Saint Michel (Fig. RC1). These tests suggest that the conclusions of larger extreme significant wave heights due to the inclusion of the water level variations occurring at high tide are still qualitatively valid. Nevertheless, the impact is larger with the coastal

wave breaking activated, which is expected to be even more significant at the end of the century with the mean sea level rise.

[Figure]

**Figure RC1: Same as Fig. 12a of the paper but with new simulations that include the depth-induced coastal wave breaking: IBI-CCS-WAV_ssh_cwb and IBI-CCS-WAV_cwb.**

Moreover, the limitations associated with the use of parameterizations have been moved to the Discussion section (see comment below). The limitations of the study are also reminded in the Abstract and in the Conclusion.

A sentence has been added at the end of the Abstract: "However, as the wave setup is computed with a parameterization based on offshore characteristics, the depth-induced wave breaking is not activated in the model. The estimates provided in this study therefore only partially represent the processes responsible for the wave-water level non-linear interactions."

The end of the Conclusion has been revised: "However, as the regional wave model does not have a very fine representation of the bathymetry or the coastline and does not resolve the depth-induced wave breaking in the shallow areas, the estimates provided in this study only partially represent the processes responsible for the wave-water level non-linear interactions. Moreover, the results found might be dependent on the parametrization used to compute the wave setup and therefore on the beach slopes.

**Specific comments**

**Section 2 Methods: model and simulations:** I find this section hard to read. I appreciate that all information required on the models are provided in section 2, and figure 1 is helpful to understand the simulation used, however it is easy to get lost in the nomenclature of the multiple simulations, and in the mere amount of information laid out. It may be worth considering simplifying the reader's work by adding a table containing a list of all simulations ran, including which forcing were used and the main details (resolution, period etc.) for each. This would improve the readability.

The structure of "Sect. 2 Methods" has been totally revised according to the suggestions of RC1. It should be now easier to read. With these changes, Figure 1 presenting all the simulations (periods, resolutions, names), appears at the very beginning of the methodology section, so we considered it would not be necessary to add a table providing the same information.

**Line 217-223:** 'Therefore, at first order, wave setup and runup can be predicted via empirical formulations […] wave setup estimates are based on an empirical formulation (Stockdon et al., 2006).' This is the paragraph where you should convince the reader that runup estimation is reliable, despite the model's approximation. Please give more details on the parametrization limitations and how the empirical formulation you use affects results (i.e. what processes you are missing out). Explain why you think this first order approximation is good enough, even though you are not including sallow water processes.

The limitations associated with the use of parameterizations (L229-237) have been moved to the Discussion section and completed by:

**"5.2 Limitations associated with the use of parametrizations for the wave setup**

Limitations in the use of parametrizations to estimate wave setup are thoroughly discussed in Melet et al., 2020; Lambert et al., 2020, including sensitivity analyses of the wave setup and runup contributions to different empirical parametrizations. The generic parametrization of Stockdon et al., 2006 used to compute the wave setup in our study is indeed subject to intrinsic limitations. A major limitation is that the formulation is only representative of sandy beaches. Other parameterizations (Guza and Thornton, 1981; Holman, 1986; dissipative case of Stockdon et al., 2006 or for a review Dodet et al. 2019) exist but they are often limited to specific coastal environments (e.g. dissipative sandy beaches, rocky cliffs) and have been calibrated with relatively few field data. The calibration therefore does not cover all the spectra of the environmental conditions.

For our large-scale study, another major limitation is that the parameterization relies on the specification of a beach slope. [L229-237]".

**Line 243:** When referring to the wave setup scaling the author sometimes use (eg. Line 243) and sometimes don't use (eg. Figur10 description, Line 433) the delta sign. Please be consistent with it.

The delta has been removed.

**Line 259-261** 'The ability of IBI-CCS-WAV and IBI-CCS-WAV_ssh to reproduce observed distributions is assessed for the mean state and the 99th percentile of the significant wave height and peak period since these variables are then used to compute the wave setup scaling (Sect. 3.2, 4).' Has the IBI_CCS_WAV_ssh been validated? The section title seems to imply it hasn't ('Validation and projections of IBI-CCS-WAV, without waves-sea level interactions').

The validation of the IBI-CCS-WAV_ssh simulation has been added only on the two scatter plots of Figure 4 and 5 which are the comparisons to tide gauge data.

A paragraph has been added in Sect. 3.1.1: "The comparison of IBI-CCS-WAV_ssh with the reanalysis is not relevant since the latter does not consider the forcing with hourly sea level variations. The IBI-CCS-WAV_ssh simulation is compared to the buoy data in Figure 4d,h and Figure 5,d,h. However, it is difficult to get useful information from these comparisons with buoys since they are not located at the coast. Actually, they are in areas where there is no impact of the wave-water level non-linear interactions (Sect. 4) so the performances of IBI-CCS-WAV_ssh are similar to those of IBI-CCS-WAV."

For the same reasons and also because we found that the inclusion of the water level variations on the wave model has no impact on the mean wave direction over the 1993-2014 period, we chose to not include the wave rose of IBI-CCS-WAV_ssh. A sentence is added Sect. 3.1.2: "In Figure 6, the focus is only on the IBI-CCS-WAV simulation as we found that the impact of the water level variations on the mean wave direction was negligible over the 1993-2014 period."

**Section 5. Discussion:** Important points are discussed, but I would strongly recommend adding a section on the implications of your results. For example:

The authors found an increase in the wave set up and a large impact on the wave-water level interaction in regions of extreme tidal range. In the introduction, the authors talk about coastal hazards and flooding during extreme water level to motivate the study. You cannot quantify hazards based on wave setup alone, but there is a lot to discuss. Considering that the tidal range will also be affected by sea level rise, are the regions where you predicted an increase in wave setup the same regions that are at risk from extreme wave events today? Are there other regions in the world that these finding could be relevant for (eg. Regions where the tidal range is expected to increase significantly)? The number of extreme events is also expected to increase in future, and your results show that these are periods in which the wave setup is particularly affected by the water-level changes; this could also be discussed. Which are the limitations of this study?

A section has been added in the Discussion part to discuss more deeply the results:

**"5.6 Implications of the results on extreme wave projections**

Marine flooding hazards cannot be quantified based on wave setup alone but wave setup can locally partially balance or enhance water levels at the coast (Melet et al., 2020a). Depending on the location (wave regimes, local ocean processes involved, sign of the extreme wave projected changes, amplitude of the projected changes in ocean processes), the inclusion of the non-linear interactions could thus enhance or balance the future wave extremes and may be important to consider for future flooding hazard calculations. The results presented in this study highlight that wave-water level non-linear interactions can be substantial for extreme wave height and wave setup, but are region dependent. For instance, the extreme wave projections are directly dependent on the water level variations forcing. In our case, the future water level variations and therefore a large part of the non-linear interactions are mainly associated with the mean sea level rise of about +80 cm and less so to changes in tides or storm surges. In other regions, large projected changes in tides or storm surges could impact the future waves conditions. For instance, Pickering et al., 2017 and Haigh et al., 2019 showed changes up to + 20 cm in the M2 component in the China Sea and in the Gulf of Saint Lawrence. Then, the future wave extremes could also be substantially more impacted in areas subject to larger mean sea level rise such as along the eastern coasts of the United States, in the Gulf of Mexico and in the Caribbean Sea where a rise of +1.4 m is expected at the end of the century under scenario SSP5-8.5 (Fox-Kemper et al., 2021)."

**Line 587, Impact of waves on sea level.** Please discuss what this means in relation to your results: how would you expect the impact of waves on sea level to affect your results?

A sentence has been added: "More importantly, they also reported a large contribution of wave induced processes to sea level extremes which are up to 20 % higher on the European continental shelf due to these wave processes. By considering these processes into the ocean model, as the water level would be higher, the impact on the wave model would be larger which means more wave-water level interactions."

**Line 605:** The new paragraph starts with 'However', it would be better to remove and start the sentence with 'The'.

Done.

**Section 6. Conclusion.** The main conclusion is not clear. I would rephrase it a way that answers your main aim reformulated as a question. For example, answer specifically: How is the sensitivity of historical and projected sea states for the IBI region coastlines affected by the non-linear interactions between wind-waves and water level changes, notably during extreme events?

As suggested by the reviewer, the sentence has been changed: "The aim of the present paper was: how is the sensitivity of historical and projected sea states to the non-linear interactions between waves and water level changes (tides, storm surge, mean sea level rise), notably during extreme events ? To assess this question, a regional wave model has been adapted to include the wave-water level interactions over the northeastern Atlantic for the 1950-2100 period." Moreover, the end of the Conclusion has been revised to describe more the limitations of the study.

**References**

Battjes, J. A. and Janssen, J. P. F. M.: Energy loss and set-up due to breaking random waves, Proceedings of 16th Conference on Coastal Engineering, Hamburg, Germany, 1978, 1978.

Dodet, G., Bertin, X., Bouchette, F., Gravelle, M., Testut, L., and Wöppelmann, G.: Characterization of Sea-level Variations Along the Metropolitan Coasts of France: Waves, Tides, Storm Surges and Long-term Changes, Journal of Coastal Research, 88, 10, https://doi.org/10.2112/SI88-003.1, 2019.

Fox-Kemper, B., Hewitt, H.T., Xiao, C., Aðalgeirsdóttir, G., Drijfhout, S.S., Edwards, T.L., Golledge, N.R., Hemer, M., Kopp, R.E., Krinner, G., Mix, A., Notz, D., Nowicki, S., Nurhati, I.S., Ruiz, L., Sallée, J.-B., Slangen, A.B.A., and Yu, Y.: Ocean, Cryosphere and Sea Level Change. In Climate Change 2021: The Physical Science Basis. Contribution of Working Group I to the Sixth Assessment Report of the Intergovernmental Panel on Climate Change [MassonDelmotte, V., Zhai, P., Pirani, A., Connors, S.L., Péan, C., Berger, S., Caud, N., Chen, Y., Goldfarb, L., Gomis, M.I., Huang, M., Leitzell, K., Lonnoy, E., Matthews, J.B.R., Maycock, T.K., Waterfield, T., Yelekçi, O., Yu, R., and Zhou, B. (eds.)]. Cambridge University Press. In Press. 2021

Lambert, E., Rohmer, J., Cozannet, G. L., and Wal, R. S. W. van de: Adaptation time to magnified flood hazards underestimated when derived from tide gauge records, Environ. Res. Lett., 15, 074015, https://doi.org/10.1088/1748-9326/ab8336, 2020.

Melet, A., Meyssignac, B., Almar, R., and Le Cozannet, G.: Under-estimated wave contribution to coastal sea-level rise, Nature Clim Change, 8, 234–239, https://doi.org/10.1038/s41558-018-0088-y, 2018.

Melet, A., Almar, R., Hemer, M., Cozannet, G. L., Meyssignac, B., and Ruggiero, P.: Contribution of Wave Setup to Projected Coastal Sea Level Changes, Journal of Geophysical Research: Oceans, 125, e2020JC016078, https://doi.org/10.1029/2020JC016078, 2020.

Pickering, M. D., Horsburgh, K. J., Blundell, J. R., Hirschi, J. J.-M., Nicholls, R. J., Verlaan, M., and Wells, N. C.: The impact of future sea-level rise on the global tides, Continental Shelf Research, 142, 50–68, https://doi.org/10.1016/j.csr.2017.02.004, 2017.

Stockdon, H. F., Holman, R. A., Howd, P. A., and Sallenger, A. H.: Empirical parameterization of setup, swash, and runup, Coastal Engineering, 53, 573–588, https://doi.org/10.1016/j.coastaleng.2005.12.005, 2006.

---

## Author Response (AR2)

24.02.2023

Author's response:

Dear editor, dear reviewers,

We finally decided to re-run the simulations with the coastal depth-induced wave breaking included for the SSP5-8.5 scenario. For the answers to the main comments, the first reviews we sent are no longer relevant. Please find in blue the answers to your main reviews taking into account the new version of the manuscript.

Best regards,

The authors.

**RC1:**

**Main comments:**

1. Methodology:

- As stated by the authors (L60-61), "wave characteristics used to estimate wave setup are sensitive to water level changes in shallow waters, where waves interact with the ocean bottom." From section 2.2.4 I understand that the wave model considers water-level variations only for the wave propagation (i.e., group velocity and wave number) while "coastal (depth-induced) breaking is not included" in the model (L92-95). My concern is that by not including the depth-induced wave-breaking the authors are missing a fundamental depth-dependent process, which can have a first order effect on the wave statistics in shallow water and hence on the wave setup. In addition, as the authors also explain in the introduction (L35-36), the wave setup is in fact due to the depth-induced wave breaking. So what I cannot understand is how the authors can assess the impact of water-level variations on the wave setup if the leading order physical mechanism driving the wave setup is not included in the model. I think the authors should carefully address this point in their manuscript.

The simulations for the SS5-8.5 scenario have been re-run with the coastal depth-induced wave breaking following Battjes and Janssen, 1978 formulation. In consequence, major changes have been made on the manuscript:

- The analyses of the SSP1-2.6 scenario have been removed
- The analyses on the wave setup have been removed and the focus in only on the significant wave height $H_s$ and peak period $T_p$. Indeed, the wave setup was calculated with a parameterization based on offshore parameters $H_s$ and $T_p$ that cannot be considered as deep-water when the coastal wave breaking is included.

- When assessing the impact of water-level forcing on the wave setup at a domain level the authors report an impact in few coastal locations. My doubt is: how much can we trust these results given the 6m minimum depth approximation?

Sect 2.3 has been revised to better explain the constraints on the bathymetry with a figure added: "For IBI-CCS-WAV, a minimum water depth of 6 meters was chosen to be consistent with that applied in the forcing from the regional ocean simulations. In the regional ocean model, this value avoids the occurrence of uncovered banks in macro-tidal areas, especially around Mont-Saint-Michel in France and in the Bristol Channel. For IBI-CCS-WAV, since the depth is set to a minimum of 6 meters (Sect. 2.2), values less than 6 meters in the tables are not used. For IBI-CCS-WAV_ssh, since the local depth fluctuates around that of IBI-CCS-WAV, values less than 6 meters in the tables are used, for example at low tide. In this case, the values can be used up to a minimum of 3 meters which corresponds to the first term of the geometric series used to discretize the tables (Fig. 2)."

Sect. 5.1 has also been revised to discuss limitations, including those associated with the minimum depth:

"**5.1 Model limitations**

The use of a single forcing climate model does not allow to quantify the uncertainties of the projected changes. Here, the focus of the study is not providing a likely range of wave projected changes over the IBI domain but rather the focus is process-oriented. In our study, the estimation of the impact of including hourly sea level variations in

the wave model is limited by several resolution aspects. The first limitation is the horizontal resolution of the wave model. The model resolution of 1/10 ° (~10 km) is conditioned by the computational cost due to the length of the simulations needed to address the question of extremes on climate scales. It does not allow a very fine representation of the coastline and of the bathymetry in the coastal zones. Moreover, the regional ocean model (Tab. 1 and Appendix A) used for the surface currents and sea level forcing does not allow for dry areas. Therefore, a minimal bathymetry is set to 6 m to run the ocean model with tides (Chaigneau et al., 2022). We chose to apply the same minimal bathymetry of 6 m in the regional wave model to maintain consistency between both regional ocean and wave models. In fact, because it would have been unrealistic to have a bathymetry of 1 m within a 10 km grid point, the minimum bathymetry (6 m) also allows to maintain a realistic balance between the 10 km horizontal resolution and the water depth. This results in fewer areas of shallow and intermediate water in the wave model and thus less effect of sea level variations on the waves. The implementation of "wetting and drying" (O'Dea et al., 2020) allowing for dry areas in NEMO version 4.2 should improve this limitation on the ocean model and therefore on the wave model. Another limitation is the resolution of the atmospheric forcing from the global climate model (Tab. 1). Given that winds are the major drivers of extreme wave events in our study, even with a relatively high-resolution climate model forcing, the resolution of 50 km for the atmospheric drivers implies that generated waves are more representative of a large-scale forcing than of coastal processes.

For all these reasons, the estimates provided in this study only partially represent the processes responsible for the non-linear interaction of sea level on waves and the results found in this study are not representative of any purely local situation at the coast but rather provide regional information. A second step of dynamical downscaling at higher resolution would be necessary to overcome such resolution limitations."

- Also, could it be that the authors found a generally small (very few locations) impact because the depth-induced breaking is neglected and the minimum depth approximation is applied?

With the coastal depth-induced wave breaking included, the impact of the sea level variations on waves is indeed larger than in the older version of the manuscript.

2. Validation:

- The title of section 3 is "Validation and projections of IBI-CCS-WAV, without wave-water level interactions" and in fact the figures of this section report data only for IBI-CCS-WAV. However, at L259-261 the authors state that "The ability of IBI-CCS-WAV and IBI-CCS-WAV_ssh to reproduce observed distributions is assessed for the mean state and the 99th percentile of the significant wave height and peak period since these variables are then used to compute the wave setup scaling". Is the IBI-CCS-WAV_ssh validated as well? If not (as I believe is the case), then I think the authors should also include the validation for the experiment using water-level forcing since, apart from the impact on the wave setup, it is also interesting and useful to know for the wave modelling community whether including this forcing can help to improve the accuracy of the model.

The validation of the IBI-CCS-WAV_ssh simulation has been added only on the two scatter plots of Figure 4 and 5 which are the comparisons to wave buoys. A sentence has been added before Sect. 3.1.1: "In this section, it is rather IBI-CCS-WAV which is validated against the reanalysis because as IBI-CCS-WAV, the reanalysis does not consider hourly sea level variations as a forcing."

A sentence has also been added in Sect. 3.1.1 : "The IBI-CCS-WAV_ssh simulation is compared to the buoy data in the scatter plots (Fig. 3d,h) and Fig. 4d,h) but the performance of IBI-CCS-WAV_ssh is similar to that of IBI-CCS-WAV since the buoys are mostly located in deep waters (Sect. 4)." and in Sect. 3.1.2: "The focus is only on the IBI-CCS-WAV simulation since the two buoys are located in deep waters (Fig. 1a)."

3. Manuscript structure:

Thank you very much for this comment which has significantly improved the structure of the manuscript. Sect. 2 has been completely revised. To reduce the length of the paper, we moved some information from the downscaling methodology in an Appendix A as suggested by the editor, notably all the part on the external forcings (global climate forcing, wave forcing, ocean forcing). A table has been added in Sect. 2 to better explain the different simulations.

- In general, I think the structure of the paper should be substantially improved before being suitable for publication. Below, a list of possible changes:

- Section 2: this section is quite confused and not logically structured in my opinion. I would first move L101-112 as in intro of Sec. 2, improving the text and Fig 2 (the colours are to weak). Then, I think the authors could

    a) describe the numerical wave model (sec 2.1), avoiding the references to global and regional simulations (e.g. L85), since I think can confuse the reader.

    b) describe the regional wave configuration IBI-CCS-WAV (sec 2.2): this is the real focus of this paper, all the other models are used to force this model in my opinion. In addition, I would move L185-190 at the beginning of this section just to state at the beginning what is the aim of this model.

    c) describe the external forcings (sec 2.3) with three subsections:

    *) Atmospheric forcing (sec. 2.3.1), describing and validating (L138-152) CNRM-CM6-1-HR model and the fields used to force IBI-CCS-WAV. Also, please avoid the acronym GCM which is typically used for General Circulation Model instead.

    *) Hydrodynamic forcing (sec 2.3.2), describing IBI-CCS and the fields used to force IBI-CCS-WAV.

    *) Wave forcing (sec 2.3.3), describing CNRM-HR-WAV and the fields used to force IBI-CCS-WAV.

    d) Inclusion of water level variations in the regional wave model: IBI-CCS-WAV_ssh (sec 2.4).

    e) Wave setup calculation (sec. 2.5): Please check the definition of the wave setup scaling – there is a delta in the definition (L243) that I think should not be there.

- Section 4: I would first describe the impact on the entire coastal domain and after on the specific locations. Also, I think the authors should clarify better what is the rational behind the choice of those two specific locations. Why not for example the Bristol channel? The tidal range there is almost as large as in Mont-Saint Michel. Also, I would rewrite Sec. 4.2 and 4.1 (which are the most important sections in my opinion), trying to discuss more in depth what is the impact and to contextualise it, maybe moderating a bit the wording (e.g., "highly impacted") which I think it is not fully reflecting the results of the authors.

Sect. 4 has been rewritten. The impact on the entire domain is provided before the specific locations. The two specific locations are chosen in France as it is our country of interest here but in terms of processes, indeed, the Bristol Channel would also have been relevant for the effects of the large tidal range.

Some insights have been added in Sect. 4.2 to explain the observed impact of sea level on waves: "In both cases, it can be pointed out that the increase in wave height occurs at high tide. These results are in agreement with Lewis et al., 2019 and Calvino et al., 2022 who both showed a significant increase in wave height at high tide at a finer scale. In Calvino et al., 2022, this impact seems to be explained mainly by the effect of bottom friction, which is less important at high tide as the water column is higher. In the case of Arns et al., 2017, waves are higher when sea level increases (e.g. at high tide) because they break closer to the shore. In our case, additional analyses would be needed to understand which process included in the model (Sect. 2.1) is the most responsible for the non-linear interaction of sea level on significant wave height."

Another section has been added in the Discussion to discuss more deeply the results:

"**5.3 Implications of the results for coastal flooding**

The results obtained in this study have shown a large impact of sea level variations on extreme significant wave heights. Wind-waves and swell contribute to extreme sea levels at the coast via wave setup and runup (Dodet et al., 2019), combined with tides, storm surges and mean sea level changes. Marine flooding hazards cannot be quantified based on wave contributions alone but these contributions can locally partially enhance sea level changes at the coast (Melet et al., 2020). Our results show that extreme significant wave heights are strongly influenced by the effect of sea level on waves in coastal areas subject to large sea level variations or on wide continental shelves.

Depending on the region (wave regimes, sign of the extreme wave projected changes, local ocean processes involved, amplitude of projected changes in local sea level), the impact of the sea level changes on waves could be important to consider for present and future flooding hazards (e.g. for threshold exceedance calculations). For instance, future waves conditions and therefore coastal flooding could be affected in areas where large changes in tides are projected such as in the China Sea and Gulf of Saint Lawrence (Pickering et al., 2017; Haigh et al., 2019). Future extreme waves could also be significantly impacted in areas subject to large relative mean sea level rise, such as along the eastern coasts of the United States, the Gulf of Mexico and the Caribbean Sea where a rise of +1.4 m is expected by the end of the century under the SSP5-8.5 scenario (Fox-Kemper et al., 2021)."

I would describe a bit better in the Conclusions and Abstract the limitations of your study.

A sentence has been added at the end of the Abstract: "The estimates provided in this study only partially represent the processes responsible for the sea level-wave non-linear interactions due to model limitations in terms of resolution and processes included."

A sentence has been added at the end of the Conclusion : "Moreover, as the regional wave model does not have a very fine representation of the bathymetry, of the coastline and does not include the feedback of waves on sea level, the estimates provided in this study only partially represent the processes responsible for the sea level-wave non-linear interactions."

**Specific comments**

- The authors may want to add some references at L170 – 173. Here I am suggesting some possible references for the North Atlantic (which is the area I am more familiar with): the Atlantic coasts are subject to very energetic events in terms of significant wave heights, wave periods and energy flows (e.g., Masselink et al. 2016, Bruciaferri et al. 2021) whereas the Mediterranean Sea and North Sea are more sheltered areas. In addition, the zone also contains very different tidal regimes with both macro and micro tidal regimes respectively in the English Channel/Celtic Sea (Valiente et al. 2018, Stokes et al. 2021) and in the Mediterranean Sea.

Some references have been added: "the Atlantic coasts are subject to very energetic events in terms of significant wave heights, wave periods and energy flows (Masselink et al., 2016; Bruciaferri et al., 2021) whereas the Mediterranean Sea and North Sea are more sheltered areas dominated by wind waves (Chen et al., 2002; Bergsma et al., 2022). In addition, the zone also contains very different tidal regimes with both macro and micro tidal regimes respectively in the English Channel/Celtic Sea (Valiente et al., 2019; Stokes et al., 2021) and in the Mediterranean Sea."

**RC2:**

**Main comments:**

I think this paper brings an interesting contribution to the field, but before publication I would like the authors to clarify the following major point:

- The model used does not include shallow water processes such as wave breaking (line. 93), and cannot represent important interactions with the seabed in shallow regions, as the minimum depth is 6m (line. 206). These shallow water processes, as stated by the authors in the introduction, are important for wave setup and set down; yet this work estimates the setup from data that exclude them. Can the estimation of the setup, calculated excluding important shallow water processes, be trusted? Is it a reliable approximation?

The analyses on the wave setup have been removed so the comments are not longer relevant.

**Specific comments**

**Section 2 Methods: model and simulations:** I find this section hard to read. I appreciate that all information required on the models are provided in section 2, and figure 1 is helpful to understand the simulation used, however it is easy to get lost in the nomenclature of the multiple simulations, and in the mere amount of information laid out. It may be worth considering simplifying the reader's work by adding a table containing a list of all simulations ran,

including which forcing were used and the main details (resolution, period etc.) for each. This would improve the readability.

Sect. 2 has been completely revised. To reduce the length of the paper, we moved some information from the downscaling methodology in an Appendix A as suggested by the editor, notably all the part on the external forcings (global climate forcing, wave forcing, ocean forcing). As you suggested, a table has been added in Sect. 2 to better explain the different simulations.

**Line 217-223:** 'Therefore, at first order, wave setup and runup can be predicted via empirical formulations […] wave setup estimates are based on an empirical formulation (Stockdon et al., 2006).' This is the paragraph where you should convince the reader that runup estimation is reliable, despite the model's approximation. Please give more details on the parametrization limitations and how the empirical formulation you use affects results (i.e. what processes you are missing out). Explain why you think this first order approximation is good enough, even though you are not including sallow water processes.

The analyses on the wave setup have been removed so the comments are not longer relevant.

**Line 259-261** 'The ability of IBI-CCS-WAV and IBI-CCS-WAV_ssh to reproduce observed distributions is assessed for the mean state and the 99th percentile of the significant wave height and peak period since these variables are then used to compute the wave setup scaling (Sect. 3.2, 4).' Has the IBI_CCS_WAV_ssh been validated? The section title seems to imply it hasn't ('Validation and projections of IBI-CCS-WAV, without waves-sea level interactions').

The validation of the IBI-CCS-WAV_ssh simulation has been added only on the two scatter plots of Figure 4 and 5 which are the comparisons to wave buoys. A sentence has been added before Sect. 3.1.1: "In this section, it is rather IBI-CCS-WAV which is validated against the reanalysis because as IBI-CCS-WAV, the reanalysis does not consider hourly sea level variations as a forcing."

A sentence has also been added in Sect. 3.1.1 : "The IBI-CCS-WAV_ssh simulation is compared to the buoy data in the scatter plots (Fig. 3d,h) and Fig. 4d,h) but the performance of IBI-CCS-WAV_ssh is similar to that of IBI-CCS-WAV since the buoys are mostly located in deep waters (Sect. 4)." and in Sect. 3.1.2: "The focus is only on the IBI-CCS-WAV simulation since the two buoys are located in deep waters (Fig. 1a)."

**Section 5. Discussion:** Important points are discussed, but I would strongly recommend adding a section on the implications of your results. For example:

The authors found an increase in the wave set up and a large impact on the wave-water level interaction in regions of extreme tidal range. In the introduction, the authors talk about coastal hazards and flooding during extreme water level to motivate the study. You cannot quantify hazards based on wave setup alone, but there is a lot to discuss. Considering that the tidal range will also be affected by sea level rise, are the regions where you predicted an increase in wave setup the same regions that are at risk from extreme wave events today? Are there other regions in the world that these finding could be relevant for (eg. Regions where the tidal range is expected to increase significantly)? The number of extreme events is also expected to increase in future, and your results show that these are periods in which the wave setup is particularly affected by the water-level changes; this could also be discussed. Which are the limitations of this study?

A section has been added in the Discussion to discuss more deeply the results:

**"5.3 Implications of the results for coastal flooding**

The results obtained in this study have shown a large impact of sea level variations on extreme significant wave heights. Wind-waves and swell contribute to extreme sea levels at the coast via wave setup and runup (Dodet et al., 2019), combined with tides, storm surges and mean sea level changes. Marine flooding hazards cannot be quantified based on wave contributions alone but these contributions can locally partially enhance sea level changes at the coast (Melet et al., 2020). Our results show that extreme significant wave heights are strongly influenced by the effect of sea level on waves in coastal areas subject to large sea level variations or on wide continental shelves. Depending on the region (wave regimes, sign of the extreme wave projected changes, local ocean processes involved, amplitude of projected changes in local sea level), the impact of the sea level changes on waves could be

important to consider for present and future flooding hazards (e.g. for threshold exceedance calculations). For instance, future waves conditions and therefore coastal flooding could be affected in areas where large changes in tides are projected such as in the China Sea and Gulf of Saint Lawrence (Pickering et al., 2017, Haigh et al., 2019). Future extreme waves could also be significantly impacted in areas subject to large relative mean sea level rise, such as along the eastern coasts of the United States, the Gulf of Mexico and the Caribbean Sea where a rise of +1.4 m is expected by the end of the century under the SSP5-8.5 scenario (Fox-Kemper et al., 2021)."

**Line 587, Impact of waves on sea level.** Please discuss what this means in relation to your results: how would you expect the impact of waves on sea level to affect your results?

A sentence has been added: "More importantly, they also reported a large contribution of wave induced processes to sea level extremes which are up to 20 % higher on the European continental shelf due to these wave processes. By taking these processes into account in the ocean model, as the sea level would be higher, the impact on the wave model would be larger, meaning an increase in waves-sea level feedbacks."

**Section 6. Conclusion.** The main conclusion is not clear. I would rephrase it a way that answers your main aim reformulated as a question. For example, answer specifically: How is the sensitivity of historical and projected sea states for the IBI region coastlines affected by the non-linear interactions between wind-waves and water level changes, notably during extreme events?

The section has been rewritten.

**References**

[revised manuscript text omitted]

---

## Author Response (AR3)

25.04.2023

Author's response:

Dear editor, dear reviewers,

Thank you for the reviews. Please find in blue the answers to your reviews taking into account the new version of the manuscript.

Best regards,

The authors.

**RC1:**

1) My main point is sec 2.1. I think this Sec. is not clear enough about the physical processes included or not in the model. In particular, in this version of the paper the authors claim that "coastal depth-induced breaking that occurs in shallow waters is parametrized using Battjes and Janssen, 1978" (L78-79). This choice is in line with current state-of-the-art large scale spectral wave models - e.g. UK Met Office wave configurations (Valiente et al. 2022). However, in the first version of the paper the authors stated: "MFWAM primarily aims at describing the open ocean sea states. As such, coastal (depth-induced) breaking is not included in MFWAM" (L92-93). So I don't really understand whether shallow water dissipative processes are included or not in the model. One think that makes me think is that I also checked Law-Chune et al, 2021 since the authors suggest checking this for the technical details. However, also in that paper there is no mention about shallow water physics included or not in the model. Could the authors please clarify this point, specifying all the processes that the model takes into account and which formulation/parameterization are used? I understand these are rather technical details but they can help a lot clarifying what the model should be able to reproduce.

In the first version of the paper, the coastal depth-induced wave breaking was not included and this process is also not included in Law-Chune et al., 2021 as you mentioned. For the second version of the paper, since we are interested in the coastal shallow waters processes, we additionally include (re-run the simulations) in the model the dissipation due to coastal depth-induced wave breaking with the parametrization of Battjes and Janssen, 1978. The paragraph has been revised to explain that this is a special addition for our paper.

2) Sec. 2.3 is not very clear in my opinion – I really like the idea behind this section, since explains the methodology and can help others to replicate the method, but as it is written is very confused. In particular, the authors should/could think to use better Fig. 2, perhaps describing it in the text. I suggest also if possible to use a consistent mathematical notation – e.g. I think it shouldn't be $k\_0$ or $k\_1$ but instead $k(t=0)$ and $k(t=1)$ (or just $k(0)$ and $k(1)$) … Anyway, with the current text I really didn't understand how the computations with the look-up tables are done.

The section has been revised. The details of the implementation of sea level changes in the wave model (e.g. look-up tables, Figure 2) have been revised and moved in a technical appendix to simplify the text for the readers. However, I didn't replace $k\_0$ by $k(t=0)$ or $k(0)$ as this is not the time but the vertical discretization in meters. I replaced $k\_0$ by $z\_0$ which is more meaningful for this variable.

3) Sec. 2.4 I think should summaries a bit the method used for the extreme value analysis.

done.

4) L166-167: I would rewrite this sentence, make it clearer that IBI-CCS-WAV_ssh is compared only against buoy and why.

done.

5) L323-326: please rephrase this sentence since I really don't understand what the authors are trying to say – e.g. 70% where does it come from?

The sentence has been revised: "In the southern North Sea, projected changes in both significant wave height and peak period are small (<10 cm, Figure 6a). The small impact of the non-linear interaction of sea level on waves (+3 cm, + 0.05 s) is therefore not negligible."

**Editor :**

- Abstract still refers to M-St-M tidal range of 10m (it is more than this). Please check.

"in average" has been added.

- Fig 1b colorbar would make more sense labelled as "depth difference" or "M2 tidal range"
- Similarly Fig 3 right hand colour bars should be labelled as difference in T, difference in H
- Same on Fig4.
- Fig 8/9 are superficially similar but very different in content. To help readers comparing parts of the paper I suggest titling them "Mean state" and "Extreme condition"
- Fig 10: To improve accessibility please add a key with line colours.
- I agree with Reviewer 2 that section 2.3 is a little confused from about line 133 and needs the wording tidying up. The bit about discretisation and the yellow arrows on the figure are simply "depth including tide is rounded to the nearest 15cm and truncated to a minimum depth of 3m" I think? Whilst depth in the tide-less model is truncated to 6m?

This is true at the surface. But, as the depth discretization follows a geometric series, the depth discretization is not constant (not always 15 cm). The deeper you get, the coarser the discretization (more than 15 cm). A sentence has been included for the deeper regions.

**References**

Battjes, J. A. and Janssen, J. P. F. M.: Energy loss and set-up due to breaking random waves, Proceedings of 16th Conference on Coastal Engineering, Hamburg, Germany, 1978, 1978.